# Dynamic Linear Modeling estimates of long-term ozone trends from homogenized Dobson Umkehr profiles at Arosa/Davos, Switzerland

Eliane Maillard Barras[1], Alexander Haefele[1], René Stübi[1], Achille Jouberton[2], Herbert Schill[3], Irina Petropavlovskikh[4,5], Koji Miyagawa[5], Martin Stanek[6], and Lucien Froidevaux[7]

[1]Federal Office of Meteorology and Climatology, MeteoSwiss (MCH), Switzerland
[2]now at Swiss Federal Institute for Forest, Snow and Landscape Research (WSL), Birmensdorf, Switzerland
[3]Physikalisch-Meteorologisches Observatorium Davos, World Radiation Center, Switzerland
[4]Cooperative Institute for Research in Environmental Sciences (CIRES), University of Colorado, Boulder, CO, USA
[5]National Oceanic and Atmospheric Administration (NOAA), Global Monitoring Lab, Boulder, CO, USA
[6]Solar and Ozone Observatory, Czech Hydrometeorological Institute, Hradec Kralove, Czech Republic
[7]Jet Propulsion Laboratory, California Institute of Technology, Pasadena, CA, USA

**Correspondence:** Eliane Maillard Barras (eliane.maillardbarras@meteoswiss.ch)

**Abstract.**

Six collocated spectrophotometers based in Arosa/Davos, Switzerland, have been measuring ozone profiles continuously since 1956 for the oldest Dobson instrument and since 2005 for the Brewer instruments. The datasets of these two ground-based triads (three Dobsons and three Brewers) allow continuous intercomparisons and derivation of long-term trend estimates. Mainly, two periods in the post-2000 Dobson D051 dataset show anomalies when compared to the Brewer triad time series: in 2011-2013, an offset has been attributed to technical interventions during the renewal of the spectrophotometer acquisition system, and in 2018, an offset with respect to the Brewer triad has been detected following an instrumental change on the spectrophotometer wedge.

In this study, the worldwide longest Umkehr dataset (1956-2020) is carefully homogenized using collocated and simultaneous Dobson and Brewer measurements. A recently published report (Garane et al., 2022) described results of an independent homogenization of the same dataset performed by comparison to the Modern-Era Retrospective analysis for Research and Applications version 2 (MERRA-2) Global Modeling Initiative (M2GMI) model simulations. In this paper, the two versions of homogenized Dobson D051 records are inter-compared to analyze residual differences found during the correction periods. The Aura Microwave Limb Sounder (MLS) station overpass record (2005-2020) is used as an independent reference for the comparisons. The two homogenized data records show common correction periods, except for the 2017-2018 period, and the corrections are similar in magnitude.

In addition, the post-2000 ozone profile trends are estimated from the two homogenized Dobson D051 time series by Dynamical Linear Modeling (DLM) and results are compared with the DLM trends derived from the colocated Brewer Umkehr time series. By first investigating the long-term Dobson ozone record for trends using the well-established multi-linear regression (MLR) method, we find that the trends obtained by both MLR and DLM techniques are similar within their uncertainty ranges in the upper and middle stratosphere but that the trend's significances differ in the lower stratosphere. Post-2000 DLM trend estimates show a positive trend of 0.2 to 0.5 %/year above 35 km, significant for Dobson D051 but lower and therefore

non significantly different from zero at the 95% level of confidence for Brewer B040. As shown for the Dobson D051 data record, the trend seems to become significantly positive only in 2004. Moreover, a persistent negative trend is estimated in the middle stratosphere between 25 and 30 km. In the lower stratosphere, the trend is negative at 20 km with different levels of significance depending on the period and on the dataset.

# 1 Introduction

The stratospheric ozone layer is essential for its role in protecting the Earth's surface from harmful solar ultraviolet radiation. Stratospheric ozone depletion occurring during the second half of the twentieth century has been contained by the strict application of the Montreal Protocol and its amendments (MontrealProtocol, 1987). While in the upper stratosphere (10–1 hPa, 32–48 km), ozone has started to show significant signs of recovery (e.g. Petropavlovskikh et al., 2019), in the lower stratosphere (147–32 hPa, 13–24 km), measurements show that ozone is still decreasing (Ball et al., 2018). Uncertainties remain for the middle stratospheric trends (32–10 hPa, 24–32 km) with different composites showing different changes giving a picture of a relatively flat trend with low significance (Ball et al., 2018).

Intensive discussions about the significance of the lower stratospheric trends and about the discrepancies between the magnitudes of the model simulated and the measured ozone trends are ongoing in the recent literature. Chipperfield et al. (2018) points to large interannual variability rather than an ongoing downward trend. Wargan et al. (2018) confirms the negative trend in the lower stratosphere in the northern hemisphere (NH) using DLM on MERRA-2 reanalysis. Sensitivity analyses by Ball et al. (2019) and Dietmüller et al. (2021) support the negative NH lower stratosphere trends highlighting, for the former, the overestimated magnitude of the final years (-2018) anomalies by the models and, for the latter, the underestimated probability density function of the model trends, as causes for the bad accordance between the simulated and measured ozone lower stratospheric trends. Orbe et al. (2020) associates the negative NH lower stratospheric trends with a change in advection, describing a northward upwelling expansion associated with an enhancement of the downwelling over NH mid-latitudes. In this case, the discrepancies in magnitude between the lower stratospheric trends retrieved from the measurements and from the models (M2GMI and MERRA-2 reanalysis) are attributed to an imperfect simulation of the tropical convective processes and of the 2016 inversion of the QBO.

Multilinear regression (MLR) is widely and consistently used for vertically resolved ozone trend estimation. This is the dominant method in the recent and past trend estimates literature (e.g. Petropavlovskikh et al., 2019; Sofieva et al., 2021; Maillard Barras et al., 2020; McPeters et al., 1996b; Reinsel et al., 2002; Tummon et al., 2015; WMO, 1998; Staehelin et al., 2001, and references therein). Trend estimates are obtained by fitting a MLR function to the monthly mean ozone time series, presuming a linear dependence of the ozone content towards the explanatory variables and a linear increase or decrease of the ozone content over time. Upper stratospheric post-2000 ozone trends are reported to be significantly positive in the three broad latitude bands, with values of $\sim 2.2\pm0.7\%$ per decade at 2.1 hPa in the NH, while non-significant negative ozone trends are derived in the lowermost stratosphere, with however large uncertainties (Godin-Beekmann et al., 2022).

The sensitivity of the post-2000 trend magnitude to the start and end years has been extensively discussed (e.g. Petropavlovskikh et al., 2019; Bernet et al., 2019; Dietmüller et al., 2021). Non monotonic post-2000 trends are also reported in Arosio et al. (2019) where MLR trends are estimated from a merged SCIAMACHY (SCanning Imaging Absorption spectroMeter for Atmospheric CHartographY), OMPS (Ozone Mapping and Profiler Suite) and SAGE (Stratospheric Aerosol and Gas Experiment) II dataset on the 2003 to 2018 period. In their study, stratospheric tropical trends are shown to be negative during the 2004 to 2011 period and positive since 2012.

Trend estimates by DLM are recent in the literature. First reports are from Laine et al. (2014) who developed the DLM analysis for trend evaluation and applied it to a merge of SAGEII and GOMOS (Global Ozone Monitoring by Occultation of Stars) data records. They compare trend estimates by DLM to trend estimates by piecewise MLR, the latter being described in a companion paper by Kyrölä et al. (2013). They conclude that DLM is a robust method well suited for modeling ozone time series changes (see Section 4.2). Their results show a statistically significant turnaround in the ozone time series after 1997 at mid-latitudes in the 35 to 55 km altitude range and a more complex behavior of the ozone concentration than the description which can be made by a simple piecewise multilinear regression model. Consequently, stronger ozone variations (decrease or increase) are reported locally when estimated by DLM than by MLR. Ball et al. (2017) applied DLM on a Bayesian composite (BASIC - BAyeSian Integrated and Consolidated) of satellite data records. The changes in ozone between 1998 and 2012 estimated using DLM indicate a clear and significant ozone recovery in the upper stratosphere. DLM has also been used to estimate trends in the lower stratosphere based on the merged SWOOSH/GOZCARDS (Stratospheric Water and Ozone Satellite Homogenized/Global OZone Chemistry And Related Datasets for the Stratosphere) data records (Ball et al., 2018) as discussed previously. More recently, DLM trend estimates on SOS (SAGEII, Osiris (Optical Spectrograph and InfraRed Imaging System) and SAGEIII) merged satellite data record are reported (Bognar et al., 2022) and indicate a clear upper stratospheric ozone recovery with varying turnaround years depending on the latitude, a decrease since 2012 in the NH upper/middle stratosphere, but without excluding a step in the Osiris dataset as a cause, and a persistent decrease in the tropical lower stratosphere.

Dobson Umkehr ozone profile data records, which are distributed all around the world (Petropavlovskikh et al., 2022; Godin-Beekmann et al., 2022; Stone et al., 2015; Miyagawa et al., 2009; Garane et al., 2022), have been extensively used in the pre-1998 stratospheric trend estimates (Reinsel et al., 1989; Randel et al., 1999; Miller et al., 1995) . Beginning in 1956 for the oldest, the Umkehr records were unique at that time since satellites records only became available in 1979 (McPeters et al., 1996a; Bhartia et al., 2013) and ozonesondes, starting in 1960 (Smit et al., 2007), do not reach the upper stratosphere. Few studies based exclusively on Umkehr measurements report on NH post-2000 stratospheric ozone trends (Zanis et al., 2006; Park et al., 2013). Zanis et al. (2006) derived trends from the Arosa Dobson Umkehr dataset and reported statistically significant negative trends in the 1970 to 1995 period, and the first signs of a reversing trend in the lower and the upper stratosphere for the period 1996 to 2004. Since this turnaround was not statistically significant, the authors suggested that the dataset should be reevaluated at a future stage when more measurements become available. The homogenized Umkehr time-series was used by Park et al. (2013) to derive trends using functional mixed models, and in the frame of the LOTUS project (Petropavlovskikh et al., 2019), which derived stratospheric ozone trends from improved and combined datasets (satellites, ground-based and

models). The NH trends derived from the Umkehr datasets are in accordance with trends derived from other ground-based instruments for the pre-1997 period and the post-2000 period. Umkehr data corroborate also the satellite findings showing highly statistically significant evidence of declining ozone concentrations since the mid 1980s in the upper stratosphere and post-2000 positive trends ranging between 2.0% and 3.1% per decade in the upper stratosphere of NH mid-latitudes. The Umkehr data records are still extensively used for trend estimates along with datasets from other ground based techniques,

satellites and models (Steinbrecht et al., 2017; Harris et al., 2015; Petropavlovskikh et al., 2019; Tarasick et al., 2019; Godin-Beekmann et al., 2022). However, trend estimations on Brewer Umkehr data records are sparse. A study using simple linear regression, without consideration of explanatory variables, applied to data from the Brewer 005 of Thessaloniki presented by Fragkos et al. (2018) reports 1997-2017 statistically significant positive trends, in the NH, above 35 $\mathrm{km}$ of 0.3%/year and non statistically significant trends below. Fitzka et al. (2004) reports on linear trends estimated with the Senn's Q method

and significances assessed with the Mann-Kendall test. We innovate here by estimating Brewer Umkehr trends considering explanatory variables in the regression by DLM.

The dataset quality is of primary importance for trend studies, and multi-instrument comparison analyses are suited to assess the long-term stability of data records by estimating the drift and bias of instruments (Hubert et al., 2016). Using microwave radiometer data records, Bernet et al. (2019) showed the effect of instrumental artefacts on the long-term ozone profile trends.

Recently, trend estimated on updated and reprocessed ozone profiles datasets have resulted in reduced trend uncertainties (Godin-Beekmann et al., 2022).

The quality of the Arosa/Davos total column ozone (TCO) dataset is currently under investigation by a reprocessing and a homogenization with the use of ozone absorption cross section from Serdyuchenko et al. (2014) (Gröbner et al., 2021) and the consideration of the effects of the relocation from Arosa to Davos (Stübi et al., 2021b). In Arosa/Davos, the Dobson

D051 is the station's primary instrument for continuous Umkehr profile time serie. It was dedicated exclusively to Umkehr measurement from 1988 until February 2013, when total ozone measurement was added to the schedule. The number of observations dedicated to Umkehr was not impacted and the number of retrieved Dobson D051 Umkehr profiles was kept to two profiles per day up to now. This frequency in observations allows the computation of statistically reliable monthly means for trend estimations. However, the instrument operations recently suffered from anomalies following technical interventions.

Therefore, a complete homogenization of the Dobson D051 Umkehr data record has been performed and is described in this paper. Trend estimations free from known instrumental artefacts can then be derived from this dataset.

The paper is organized as follows: the data sources used in this study are described in section 2, with a special focus on the Umkehr method description. In section 3, the complete homogenization of the Dobson D051 Umkehr data record is detailed and compared to the homogenization performed by NOAA on the same data record in the frame of the ESA project WP-2190

(Garane et al., 2022). The MLR and DLM trend estimate methods are described in section 4, with a comparison of the trend values resulting from both regressions on the same Dobson D051 data record. Results of vertically resolved long-term trend estimates by DLM are presented and discussed in section 5, followed by conclusions in section 6.

## 2 Data Sources

### 2.1 Umkehr data records from Arosa/Davos

The Umkehr technique, which will be described in section 2.1.1, allows low-resolution retrieval of ozone profiles from measurements made by Dobson and Brewer spectrophotometers. TCO and ozone profile measurements with Dobson (and Brewer) spectrophotometers were performed at Arosa (46.82° N, 6.95° E) from 1926 (and 1988) to 2021 and at Davos since 2012. For a detailed description of the Dobson and Brewer spectrophotometers, we refer to Stübi et al. (2021a, 2017a). The progressive relocation of the Dobson and Brewer triads from Arosa to Davos (13 km north of Arosa and 260 m lower in altitude) between

2012 and 2021 is described and analyzed in Stübi et al. (2017b, 2021b). Umkehr measurements are performed under clear sky and low cloud cover conditions twice a day since 1956 by Dobson spectrophotometers (Dobson D015 since 1956 and then Dobson D051 since 1988), 4–6 times per month by Dobson D101 since 1988 and by Dobson D062 since 1998. Dobson D051 performs fully automated Umkehr measurements since 1988. The Dobson Umkehr measurements are complemented by Brewer Umkehr measurements since 1988 with Brewer B040 and since 2005 with Brewers B072 and B156. See Table 1 for

a summary of the time ranges and time resolutions of the six Arosa/Davos spectrophotometers. Ozone profile Umkehr measurements initiated in 1956 at Arosa and continued since 2021 at Davos, Switzerland, compose the longest continuous Umkehr measurement time series world-wide (Staehelin et al., 2018).

| Instrument | | Time range | Time resolution |
|---|---|---|---|
| Dobson | D015 | 1956-1988 | 2 profiles/day |
| | D051 | 1988-now | 2 profiles/day |
| | D062 | 1998-now | 4-6 profiles/month |
| | D101 | 1988-now | 4-6 profiles/month |
| Brewer | B040 | 1988-now | 2 profiles/day |
| | B072 | 2005-now | 2 profiles/day |
| | B156 | 2005-now | 2 profiles/day |

**Table 1.** Time ranges and time resolutions of the Dobson and Brewer Umkehr measurements at the Arosa/Davos station.

At Arosa, the Dobson D051 sat on a turntable in a conditioned hut maintained at 25-28 °C. An aperture in the roof, which opened and closed according to solar zenith angle (SZA) and weather conditions, allowed zenith measurements. The continuous

and automated measurements (2 min cycle) are interpolated to 12 nominal SZAs and profiles are retrieved from ground to 50 km using Optimal Estimation Method (OEM) (Rodgers, 2000) implemented in Petropavlovskikh et al. (2005a). Manual Umkehr measurement started in 1968 with the Dobson D101 and in 1992 with the Dobson D062 as redundant measurements to check the stability of Dobson D051. These are made two to three times each month since 1988 and 1998 in favourable weather conditions. The Dobson D062 and the Dobson D101 have been automated in 2012 and in 2013, respectively (Stübi

et al., 2021b). They are located since 2021 in Davos in a common air conditioned container side by side with the Dobson D051, and measure Umkehr curves through a quartz dome. While the Dobson D051 was dedicated exclusively to Umkehr measurement until February 2013, the present set-up allows both direct sun and zenith Umkehr measurements with the three Dobsons. The Arosa/Davos Dobson instruments are regularly calibrated against the two European regional secondary reference Dobson instruments D064 from the Hohenpeissenberg Observatory (MOHp, Germany) and D074 from the Solar and Ozone Observatory in Hradec Králové (SOO-HK, Czech Republic) (Stübi et al., 2021b, Fig. 3).

The Brewer triad consists of two Brewer Mark II single-monochromator instruments, the Brewer B040 and the Brewer B072, and one Brewer Mark III double-monochromator instrument, the Brewer B156. The three instruments measure daily in Umkehr mode when the sun is at the 12 nominal SZAs. Since the operation of the first Brewer at Arosa in 1988, biennial calibrations have been carried out (Stübi et al., 2017a, Fig. 1) towards the traveling reference instrument Brewer B017 and, since 2008 towards the traveling reference instrument Brewer B185. The instruments of the Brewer triad underwent very few technical interventions and are in good agreement with the travelling references (TCO deviations <=1%, (Stübi et al., 2017a)). In particular, no technical issues are reported around 2011-2013 and 2018, which are data records periods considered in the frame of the Dobson D051 homogenization. However, sporadic instabilities in the Brewer B072 data record have been observed while no particular technical issues have been detected by the intercomparison procedures. The Dobson D051, the Brewer B072 and the Brewer B156 have been simultaneously relocalized from Arosa to Davos in September 2018 however with an effect on the TCO level within the instrumental noise (Stübi et al., 2017a, 2021b).

### 2.1.1 The Umkehr method

The Umkehr method is based on the measurement of the ratio of downward scattered zenith sky radiation for two wavelengths in the UVB-UVA range from 300 nm to 330 nm (Huggins absorption band) which are subject to different strengths of ozone absorption, the shorter wavelength being more strongly absorbed by ozone. This ratio changes as a function of SZA during sunset and sunrise due to changes in the scattering height along the zenith (Mateer, 1965; Stone et al., 2015). As the SZA is increasing from 60° to 90°, the scattering height is increasing, and the two intensities decrease because of increased absorption and scattering by ozone and air molecules. As the shorter wavelength has a higher scattering point than the longer wavelength, its intensity is decreasing faster than the longer wavelength intensity as long as both scattering heights are below the ozone maximum. At high SZA, the scattering height for the shorter wavelength is above the ozone maximum and the scattering height of the longer wavelength is still below the ozone maximum. The shorter wavelength intensity decreases then less rapidly than the longer wavelength intensity and the ratio reaches a maximum at high SZA called the Umkehr effect (Götz et al., 1934). The Umkehr method allows the retrieval of ozone profiles from the measurements by Dobson and Brewer spectrophotometers. We describe the particularities of Dobson and Brewer Umkehr measurements in the following subsections.

### 2.1.2 Umkehr measurements by Dobson spectrophotometer

The logarithm of the ratio of the two wavelengths intensities (R values) is converted to radiance using calibration tables (RtoN table) and reported as N values (Fig.1a) in N-units for 12 nominal SZAs between 60° and 90°(60°, 65°, 70°, 74°, 77°, 80°, 83°,

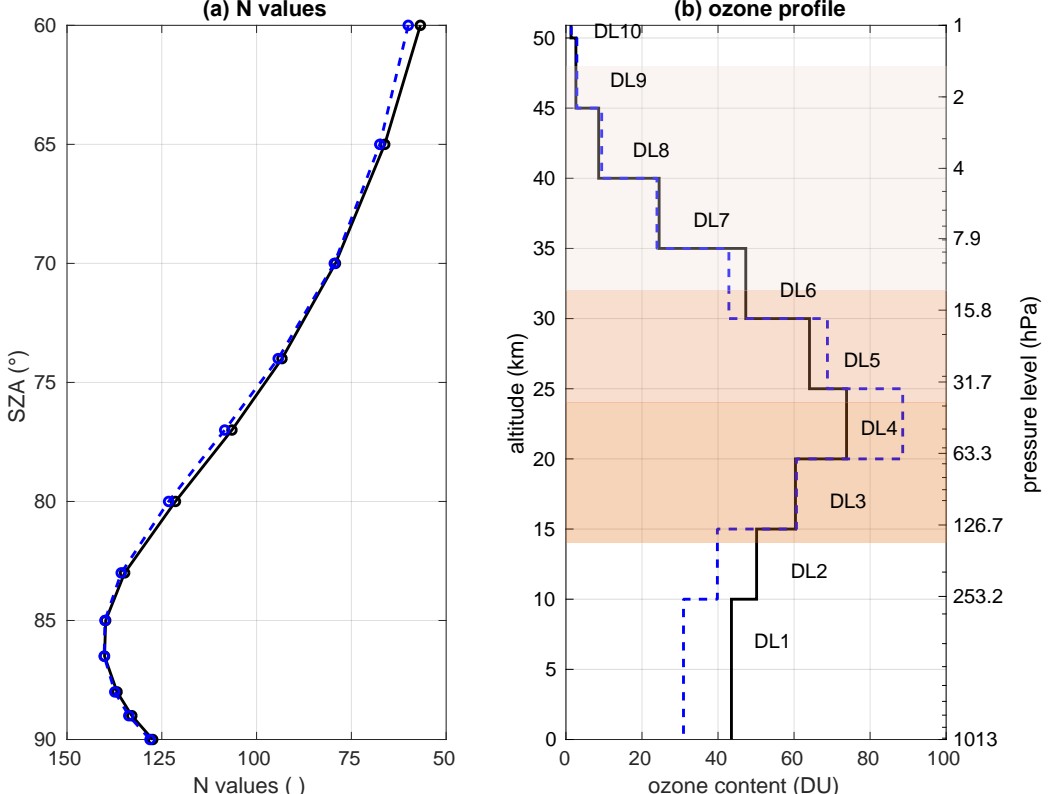

**Figure 1.** (a) Morning (in black) and afternoon (in blue) N curves at 12 nominal SZAs and (b) their corresponding retrieved ozone profiles in DU as a function of altitude in km and pressure level in hPa. Total column ozone and atmospheric conditions slightly differs between the morning and the afternoon. The altitude ranges of the 10 Dobson layers (DL) are shown in (b). Lower, middle and upper stratospheric ranges are displayed in orange shadings.

85°, 86.5°, 88°, 89° and 90°). The nominal wavelength pairs used in a Dobson spectrophotometer are A: 305.5 & 325.4 nm, C: 311.45 & 332.4 nm, and D: 317.6 & 339.8 nm. Two narrow slits separate the respective wavelengths. The ozone profiles
180  (Fig.1b) are retrieved from the measurements of the C pair intensity while the total column measurement uses a combination of two wavelength pairs (AD) (Stübi et al., 2021a).

The 12 N values (further called N curve) are screened for clear sky conditions and corrected for cloud influence using a nearby UV/VIS lux meter. This empirical correction is based on the relation between the UV/VIS intensity of clear days (within the same month, for each SZA) and the UV/VIS intensity variation during the cloudy N curve measurement (see
185  Basher, 1982). This cloud correction is based on a uniform cloud layer and may fail for more complicated cloud structures. Haze correction is not included. It was shown that the effect of small cloud corrections of the N values on the vertically resolved

ozone trends is negligible. For these reasons, only profiles retrieved from N curves without any cloud correction or with a small correction are considered for our study.

### 2.1.3 Umkehr measurements by Brewer spectrophotometer

The intensity of 8 wavelenghths (306.3, 310.1, 313.5, 316.8, 320.1, 323.2, 326.5, and 329.5 nm) are quasi-simultaneously measured for solar zenith angle changing from 60° and 90°. A holographic grating is used as dispersive element for the solar radiation passing then through narrow slits centered on the desired wavelengths. Mark II Brewer instruments use one single holographic grating and therefore only one dispersive element to separate the wavelengths. Mark III Brewer instruments are double monochromators that use two holographic gratings (Staehelin et al., 2003). The Umkehr ozone profile can be retrieved from three measured wavelength pairs (McElroy and Kerr, 1995; Stone et al., 2015) by OEM. For similarity with the Dobson Umkehr measurement, the intensity ratio of only two wavelengths are used here: 310.05 nm of short set of wavelengths, and 326.5 nm of the long set of wavelengths. The data are flagged for clouds before the interpolation onto the 12 nominal SZAs. The quality filter eliminates data points that fall outside a predefined error envelope determined by the range of natural variability and a mean offset.

### 2.1.4 Ozone profiles retrieval

Retrieved ozone profiles are given on ten layers between 0 and 50 km with a vertical resolution of 10-15 km. Dobson and Brewer ozone profiles are retrieved by OEM. The Dobson Umkehr retrieval algorithm is described in Petropavlovskikh et al. (2005a) and the Brewer Umkehr retrieval algorithm has been adapted by Petropavlovskikh et al. (2005b) from the Dobson algorithm. The version of the code used in this study has been implemented by M. Stanek and can be found at http://www.o3soft.eu/o3bumkehr.html. Dobson and Brewer Umkehr retrievals are using the same a priori profile, the ML climatology, described in McPeters and Labow (2012) formed by combining data from Aura Microwave Limb Sounding (MLS) (2004-2010) with data from ozonesondes (1988-2010). The measurement error covariance matrices are diagonal with values between 0.16 to 0.8 N-units for Dobson and 0.6 to 2 N-units for Brewer. The Brewer observation errors have been estimated by the standard deviation of the 2005–2018 climatological difference of collocated and simultaneous N values measurements. In the layers below Dobson Layer (DL) 4, peaking at 20 km, for both instruments, the Averaging Kernels (AKs, not shown) show sensitivity of observations to ozone variability in several layers, and therefore the partitioning of the retrieved ozone in individual layers is based on the a priori information.

The quality check of the retrieved ozone profile includes assessment of the number of iterations (fewer than four is considered a good profile) and the condition that the difference between observed and retrieved Umkehr observations at all SZAs remains within measurement uncertainty (Petropavlovskikh et al., 2022).

A generic stray light correction can be applied to reduce systematic biases in the Dobson Umkehr retrieved profiles (Petropavlovskikh et al., 2011). The NOAA version of the Dobson retrieval applies this correction while the MeteoSwiss (MCH) version does not. The seasonal bias between the Dobson and Brewer ozone records is reduced when a stray light correction is applied to

the Dobson record (Petropavlovskikh et al., 2009). Moreover, as a step change in the record can be related to a change in the amount of stray light, a proper correction of the stray light effect can help to reduce the magnitude of the step.

The Dobson D051 Umkehr observations dataset is regularly archived at the World Ozone and Ultraviolet Radiation Data Centre (WOUDC, www.woudc.org). The Brewers are part of the eubrewnet (http://www.eubrewnet.org/eubrewnet), where raw data files are available for registered users.

## 2.2 Aura MLS

The Microwave Limb Sounder (MLS) is a microwave limb-sounding radiometer on board the Aura Earth observing satellite, launched in July 2004. Ozone profiles are retrieved from Aura MLS radiance measurements at 240 GHz. Details about the instrument can be found in Waters et al. (2006). Ozone profiles from the version 4.2 dataset are given on 55 pressure levels from 1000 to 1e-5 hPa (Livesey et al., 2018). However, the useful vertical range for Aura MLS ozone leads us to only consider Aura MLS data from 10 to 75 km (in this range, the Aura MLS vertical resolution is about 2.5 to 4 km) for Aura MLS overpasses above Arosa ($\pm3°$ in latitude and $\pm5°$ in longitude). These ozone profiles are interpolated on the Umkehr pressure levels $p_i$ and converted to DU following Godson (1962):

$$X_{DU} = C * \bar{X} * (p_i - p_{i-1}) \tag{1}$$

with $C = 0.00079 \mathrm{DU\,hPa^{-1}\,ppbv^{-1}}$ and $\bar{X}$ the ozone mean VMR in ppbv. Approximative heights are given as in Petropavlovskikh et al. (2022).

## 3 Homogenizations of the Dobson D051 dataset

As the quality of a dataset is essential in order to estimate reliable long-term trends with uncertainties as reduced as possible, we first investigate the quality of the Arosa/Davos longest Umkehr ozone profile dataset and proceed to its detailed homogenization.

The worldwide longest Umkehr ozone profile record was recently impacted by short term anomalies due to instrumental changes and technical issues.It has been homogenized by two simultaneous but independent studies, one by the principal investigator group of the Dobson D051 instrument (further called MCH homogenization) and one by the NOAA (further called NOAA homogenization). Both homogenizations are described in sections 3.1 and 3.2 and compared in section 3.3. Details are provided in this work for the MCH homogenization while the reader is referred to Garane et al. (2022) and Petropavlovskikh et al. (2022) for details on the NOAA homogenization.

## 3.1 MCH homogenization of the Dobson D051 dataset

The Arosa/Davos Umkehr time series is composed of Dobson D015 measurements from 1956 to 1988 and Dobson D051 since then. The quality of the homogenization of the Dobson D015 to Dobson D051 transition has been ensured by one year of parallel measurements (1988) allowing an adaptation of the D015 N values to the D051 N values. For each SZA, the 1988

mean difference between the D051 and the D015 N values has been added to the D015 values. The 1956-1987 ozone profiles have then been retrieved from the Dobson D015 corrected N values. No statistical correction has been performed on the D015 ozone dataset.

We report here about the complete homogenization of the 1988-2020 Umkehr Dobson D051 time series by comparison to the datasets of the five collocated instruments (two Dobson and three Brewer spectrophotometers) on the N value level. The purpose is to detect common anomalies in the difference between Dobson D051 and each of the redundant measurements and to correct the Dobson D051 time series accordingly. However, a correction is only applied if it correlates with a technical issue reported in the metadata. If we cannot see any indication in the metadata for an instrumental drift, no correction is applied.

Figure 2 shows the time series of monthly mean ozone profile differences between Dobson D051 and the 5 collocated spectrophotometers. Only simultaneous measurements, not flagged for bad weather conditions, volcanic eruptions, and number of iterations, are considered. The relative differences of the anomalies lie within $\pm15\%$. The comparisons with the Brewer instruments show a seasonal cycle with differences slightly bigger in summer than in winter (not shown; DL6: -2% in winter and +2% in summer). A similar behavior has been found by Gröbner et al. (2021) when comparing TCO from Dobsons to Brewers. Note that the annual cycle is not visible on the representation of deseasonalized anomalies as in Figure 2 and that we consider changes when they are larger than the standard deviation of the Brewer Dobson differences.

If we focus on the post-2000 period, where several collocated and redondant measurements are available, systematic anomalies of the Dobson D051 are noticed (periods in black frames in Figure 2):

- before 2003 for the altitude range below 30 km: the Dobson D051 ozone values are higher than the values measured by the collocated instruments below 20 km and lower between 20 and 30 km

- in winter 2010 above 40 km, the Dobson D051 ozone values are higher than the values measured by the collocated instruments

- between 2011 and 2013 in most part of the altitude range, the Dobson D051 ozone values are lower than the values measured by the collocated instruments

- after 2018, the Dobson D051 ozone values are higher than the values measured by the 3 collocated Brewer instruments.

The comparison of Dobson D051 with the collocated Dobsons around 2014 and after 2018 are to be taken with caution due to the very limited number of measurements of Dobson D051 in 2014 and of Dobson D062 and Dobson D101 during these periods. Around 2014 (technical and staff transition period), many data are missing or have to be flagged because of roof opening issues. After 2018, the Umkehr measurement by Dobson D062 and Dobson D101 have been drastically reduced as priority has been given to total ozone measurements.

Table 2 summarizes the Dobson D051 problematic periods, the technical issue reported at these periods, and the time ranges and redundant datasets used for the offsets determination.

When systematic for each pair of instruments and if related to an instrumental issue, the detected Dobson D051 problematic periods are shifted according to the mean difference with the 3 Brewers or the 2 Dobsons datasets before and after the problematic periods (periods of 2 years are considered). The homogenization is performed on the raw data level (N values) and the ozone profiles are then retrieved from the corrected N values. The Dobson D051 and the Brewers stay independent from each

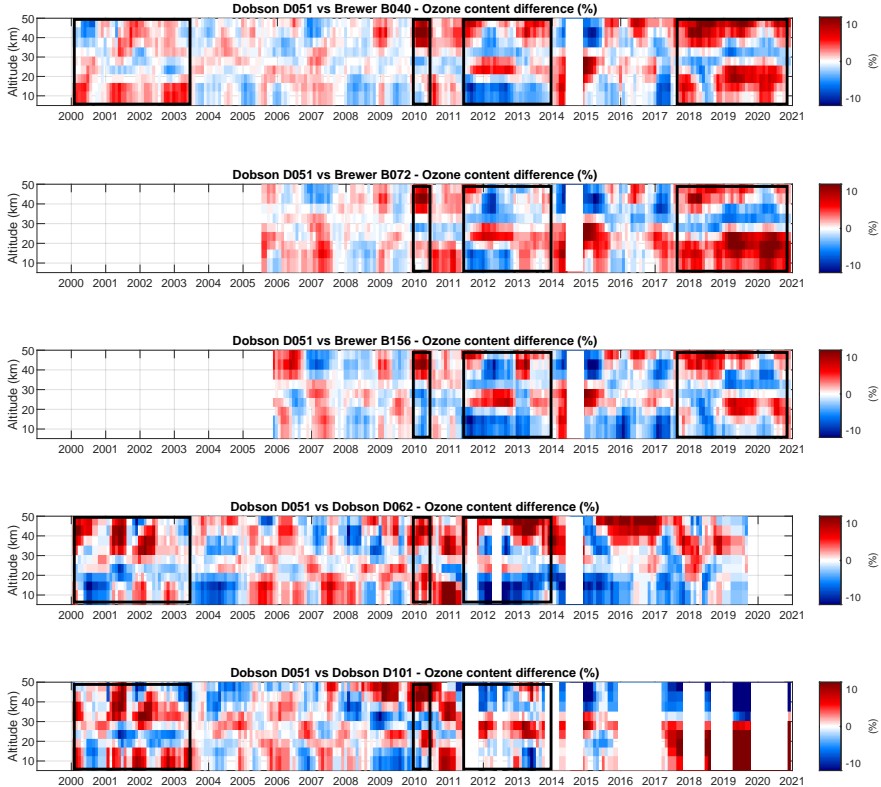

**Figure 2.** Monthly mean time series of the ozone profiles relative differences for each of the 5 spectrophotometers with respect to D051. The time series are deseasonalized and smoothed by a 6 months moving average.

other as one is not corrected to fit the ozone values of the others. Only the mean variation of the Brewers datasets during 2
285 years before and after an anomaly (Brewers data records do not suffer from anomalies during these periods) is replicated on
the same 4 years of Dobson D051, allowing the long-term ozone variations to stay independent.

For each period that requires a correction (see Table 2) we apply to the N values a SZA dependent offset which is constant
over the period to be corrected. The offset is calculated such that the difference averaged over the period and over the reference
instruments (two Dobsons in 2003 or three Brewers after 2011) matches the difference averaged over two years before and two
290 years after the period and over all reference instruments (see Fig. 3):

$$\Delta_{SZA} = mean(\Delta1_{SZA}, \Delta3_{SZA}) - \Delta2_{SZA} \tag{2}$$

$$N2_{SZA}^{corr} = N2_{SZA} - \Delta_{SZA} \tag{3}$$

| year of Dobson D051 anomaly | Technical issue/instrumental change | Homogenized period | Time range used for the offset determination | Redundant datasets used for the offset determination | Comment |
|---|---|---|---|---|---|
| 1988 | D015 to D051 | Before 1988.01.01 | 1987.01.01 - 1988.01.01 and 1988.01.01 - 1989.01.01 | D015 and D051 simultaneous measurements | Instrumental change: Dobson D051 replaces Dobson D015. Adjustement of the dataset measured by D015 before 1988 to the dataset measured by D051 after 1988 |
| 2003 | Intercomparison and new RtoN table New RtoN table considered | Before 2003.07.19 | 2001.07.19 - 2003.07.19 and 2003.07.19 - 2005.07.19 | D062 and D101 mean values | Adjustement of the optics during the IC. Remaining inhomogeneity despite the use of a new RtoN table |
| 2010 | - | 2010.01.01 - 2010.06.30 | - | - | Does not correspond to any technical issue. Period limited to 6 month. Not corrected. |
| 2011-2013 | New electronics (2011.03.21) New Qlever motors (2012.02.15) New software 3V3 (2013.03.26) | 2011.04.01 - 2013.04.01 | 2009.04.01 - 2011.04.01 and 2015.04.01 - 2017.04.01 | B040, B072 and B156 mean values | 2014 not considered (number of measurement low and problematic period). Refurbishment of the electronics (HV, motors, feedback loop, amplification board) and position of Q2-lever as function of the room Temperature. Q-lever motor are essential is the selection of the wavelengths. |
| 2018 | New wedge steel band (2018.05.06) IC(2018.08.07-17): adjustments on optics AROSA to DAVOS (2018.09.28) | Before 2018.05.01 | 2016.05.01 - 2018.05.01 and 2018.05.01 - 2020.05.01 | B040, B072 and B156 mean values | The optical attenuator consists of a moving neutral-density filter (the optical "wedge") attached to a graduated rotating disc (R dial). The wavelength pair selection is achieved by rotating a pair of quartz plates (Q1 lever, Q2 lever) through which the light beam passes. |

**Table 2.** Dobson D051 homogenization description: determined time of Dobson D051 anomaly, technical issues or instrumental change which is considered as the source of the anomaly, homogenized period, time ranges for the offset calculation, used redundant datasets for the offset calculation, and details of the technical issue

$\Delta_{\mathrm{SZA}}$ is the offset between the three Brewers mean N values and the Dobson D051 N values for each SZA, $\Delta 1_{\mathrm{SZA}}$ and $\Delta 3_{\mathrm{SZA}}$ are the difference between the three Brewers mean N values and the Dobson D051 N values before (period $\mathrm{P}_1$) and after (period $\mathrm{P}_3$) the Dobson D051 problematic period (period $\mathrm{P}_2$). All values are averaged over two years periods. $\mathrm{N2}_{\mathrm{SZA}}^{\mathrm{corr}}$ is the corrected N value in period $\mathrm{P}_2$.

In case of a step in the time series (e.g. in July 2003 and in May 2018), the period $\mathrm{P}_2$ does not exist and should not be considered in Fig. 3. The corrected N value $\mathrm{N2}_{\mathrm{SZA}}^{\mathrm{corr}}$ of period $\mathrm{P}_1$ is then obtained following equations 4 and 5.

$$\Delta_{SZA} = \Delta 1_{SZA} - \Delta 3_{SZA} \tag{4}$$

$$N1_{SZA}^{corr} = N1_{SZA} - \Delta_{SZA} \tag{5}$$

### 3.2 NOAA homogenization of the Dobson D051 dataset

In parallel but in a separate work, a homogenization and a correction for the stray light effect of the same Dobson dataset has been performed by NOAA (Garane et al., 2022; Petropavlovskikh et al., 2022). They use the comparison of the Dobson D051 dataset with the M2GMI model on the N values level when the MCH homogenization uses the comparison with N values of the collocated instruments. A summary of the homogenization method is presented here, for details on the method and for the description of the stray light correction, we refer to Petropavlovskikh et al. (2022).

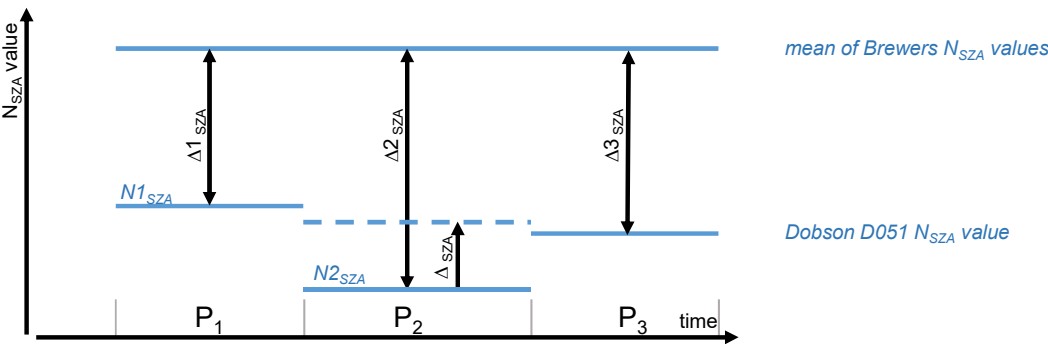

**Figure 3.** Schematic of the Dobson D051 MCH homogenization principle

The NASA Global Modeling Initiative Chemistry Transport Model (GMI CTM) (Orbe et al., 2017; Wargan et al., 2018) is a full general circulation model that is driven by MERRA2 meteorological reanalysis throught the replay method (Gelaro et al., 2017). The simulation of the meteorological fields in the M2GMI model is continuously referenced against the MERRA-2 winds, temperature and surface pressure fields (Orbe et al., 2017). For the NOAA homogenization process, the M2GMI ozone and temperature profiles are selected for the Arosa station location. The simulated temperature profile is used for accounting for the temperature dependence of the ozone cross section and allows the model to better fit to the day-to-day variability of the N values. The Umkehr retrieval forward model is using the M2GMI profiles to simulate Umkehr N values for an idealized Dobson instrument that does not have a stray light interferences. For each SZA, differences between simulated (idealized) and measured (instrument specific) Umkehr N values are averaged over the time between two consecutive calibrations (performed at each Dobson intercomparison campaign) of the Dobson D051 to create an empirical correction that accounts for the stray light of the D051 instrument. An iterative modification of the N value correction is further performed for optimization of the stray light correction, adding a constant offset correction to the Umkehr dataset. This results in a reduced bias to other ozone records in the upper stratosphere but, as a constant offset, does not have any impact on the trends. While the first iteration of the homogenization removes artificial steps in the Umkehr ozone profile records, the iterative part reduces the bias relative to other ozone observing systems.

The NOAA homogenized Dobson D051 dataset has been compared to satellites data records including AURA MLS in (Garane et al., 2022). The agreement is within $\pm$-5 % in the upper and middle stratosphere and larger biases (up to 10 %) are found in the lower stratosphere.

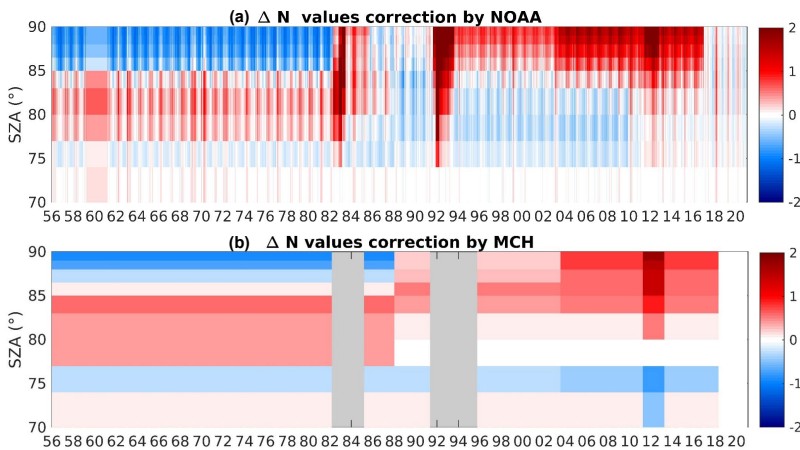

**Figure 4.** Monthly mean time series of the N values correction (a) for the NOAA and (b) for the MCH homogenization of Dobson D051 dataset. Volcanic eruptions periods (grey shaded area) are not corrected by the MCH homogenization.

### 3.3 Comparison of the homogenizations of the Dobson D051 dataset

The NOAA homogenization has been developed to remove artificial steps in the Umkehr ozone profile records and to reduce the bias relative to other ozone observing systems. The MCH homogenization approach is different in that the homogenization

process aims to remove artificial steps in the Dobson D051 Umkehr profiles record while maintaining the constant offset between the datasets, thus ensuring the independence of the Dobson D051 ozone values towards the collocated instruments datasets.

Both homogenizations provide correction offsets on the N values level and ozone profiles retrieved from the corrected N curves. We compare first the time series of the N values correction offsets. Then the homogenized ozone profiles time series

are considered by comparing the time series of their difference to Aura MLS.

Figure 4 shows the time series of the N values correction as a function of SZA as determined by the NOAA homogenization (Fig. 4a) and by the MCH homogenization (Fig. 4b, this study). For comparison purpose, the NOAA correction values have been offsetted with their mean difference after 2018.

The main differences between the two homogenizations are the variability of the corrections values and the correction of the

volcanic eruptions periods. The seasonal variability of the NOAA N values comes from the correction of observed N values for the stray light effect. Indeed, the straylight contribution varies with SZA and is proportional to the total column ozone value (Petropavlovskikh et al., 2009). For the same SZA, the amount of correction is different for each monthly mean value

of the timeseries in proportion to the seasonal changes in total column ozone (Fig. 4a). This is not corrected for in the MCH N value homogenization. The years around 1982 and 1992 are periods of volcanic eruptions (El Chichon and Pinatubo) which

are corrected by the NOAA homogenization but not considered in the MCH homogenization as the Umkehr retrieval does not account for the change in atmospheric scattering due to aerosols injection (Petropavlovskikh et al., 2022). For the 1988 to 2003 period, both homogenizations differ for the 77-83° SZAs. Otherwise, the correction amplitudes are similar, and their occurrences coincide within a few months in January 1988, in July 2003, and in April 2011 to 2013; this is remarkable given the differences in the detection method. Note that the 2010 six-months step has been chosen to be left uncorrected in the MCH

homogenization due to the absence of confirmed technical issue at that time. In 2017/2018, the start date considered for the NOAA homogenization is January 2017, while the start date considered by the MCH homogenization is May 2018 with the probable effects of the wedge steel band replacement on the measurements.

However, while both corrections of the N values look similar, small differences in the N curve shapes can lead to larger differences in the ozone profiles due to the non linear relationship between the N values and the ozone values (see the two N

curves and ozone profiles in Fig.1 for an example).

In order to evaluate the effects of both homogenization on the Dobson D051 time series, monthly mean relative difference to Aura MLS data record are plotted in Figure 5 for two altitude levels i.e. DL5 (25 km) in middle stratosphere and DL8 (40 km) in upper stratosphere. The relative difference of the Brewer B040 time series is also shown for the same layers.

The Brewer B040 relative difference shows a constant offset to Aura MLS but clear anomalies in 2012 and 2013 in DL5

(Fig.5a). The Dobson D051 homogenized by NOAA shows a very good accordance with Aura MLS both in DL5 and DL8. The small mean bias is a result of the NOAA optimization of the stray light correction. Therefore, it is not the magnitude of the bias between the homogenized dataset and Aura MLS but its variation (the bias should be constant) which should be considered here. No clear offset in the difference to Aura MLS between the NOAA and the MCH homogenized record is reported in DL5. The variability of the differences to Aura MLS of each dataset looks higher after 2010 while the mean values

are constant. However, the slight underestimation of the MCH homogenization since 2017 seems to match the Brewer B040 difference to Aura MLS in DL5 (Fig.5a). After 2017, the relative difference to Aura MLS of D051 homogenized by MCH and of the collocated B040 is within -5% to -10% while the D051 homogenized by NOAA lies within -2% of Aura MLS.

A clear correction of the 2011-2013 period is visible in DL8 (Fig.5b). Except for the respective MCH and NOAA homogenized datasets mean offsets to Aura MLS, a slight overestimation of the NOAA homogenization is visible in 2012 and 2013.

However, the Brewer B040 relative difference to Aura MLS is also slightly smaller during this time range, when the Brewer instrument had not undergone any technical interventions. This is particularly visible on the anomalies time series of B040 in Fig. 5c. As the MCH homogenization relies on the Brewer collocated datasets, it allows to take into account the local variability of the ozone DL8 content that the M2GMI model, base for the NOAA homogenization, probably does not consider. As the atmospheric processes are more homogenized in the stratosphere that in the troposphere, the M2GMI ozone profiles should be

representative of stratospheric ozone variability. Nevertheless, it is possible that other atmospheric interferences (i.e. aerosols) can impact the Dobson readings of zenith sky radiance which would also impact Brewer observations, but might not be fully included in the M2GMI simulations.

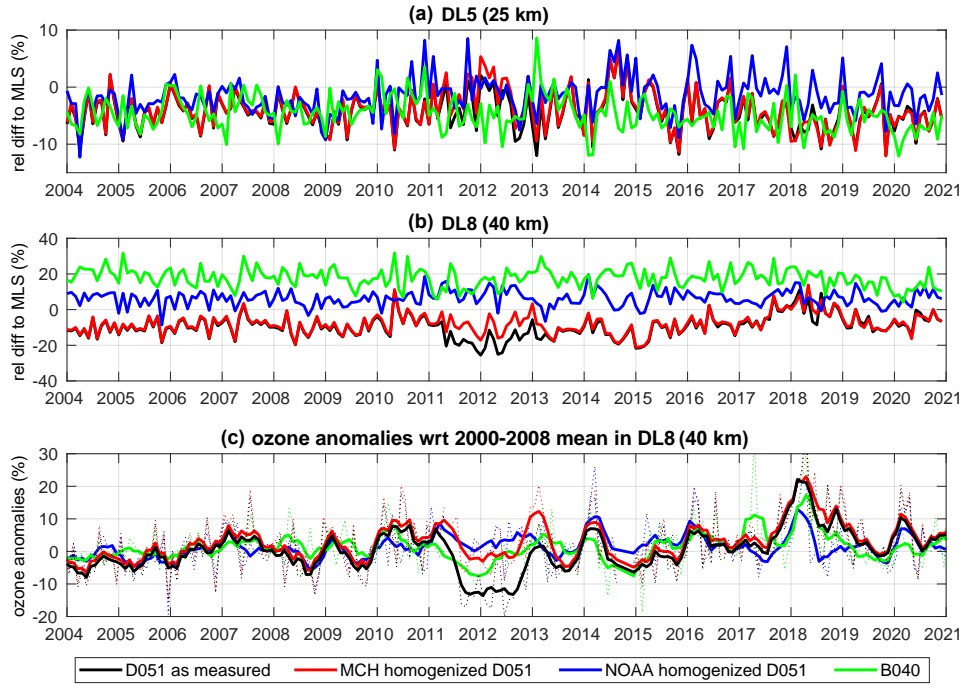

**Figure 5.** Monthly mean ozone content relative difference to Aura MLS of Dobson D051 as measured (black), Dobson D051 NOAA homogenized (blue), Dobson D051 MCH homogenized (this study, red) and Brewer B040 (green) deseasonalized time series in (a) DL5 and (b) DL8. (c) Time series of ozone anomalies towards their 2000-2008 mean for the same ground-based datasets in DL8.

Due to the occurence of an anomaly in 2018, which is particularly visible in DL8 for all datasets (Fig. 5c), the last correction applied to the dataset by the NOAA and the MCH homogenizations differ.

As the MCH homogenization considers a step correction in May 2018, the ozone increase during the 2018 anomaly is accounted for in the mean difference of the D051 dataset to the Brewers datasets of the pre- and the post-step periods. As a result, the calculated offset is small. The NOAA homogenization method detects a change in the Umkehr ozone with respect to the M2GMI record that starts a year earlier, in 2017. The ozone increase during the 2018 anomaly is accounted for only in the mean difference to M2GMI of the post-step period of the D051 dataset. Moreover, this post-step difference is overestimated as

M2GMI doesn't seem to simulate any significant anomaly at that period. As a result, the calculated offset, applied in 2017, is probably overestimated.

Now that the Dobson D051 is fully homogenized, vertically resolved long-term trends can be estimated with limited influence of instrumental artefacts.

## 4  Long term trend estimation methods

Two regression methods for trend estimation are described in this section. First, we describe the common and widely used MLR and second, we detail the more recent DLM regression method. Trends estimation by both methods are then compared on the case study of MCH homogenized Dobson D051 dataset.

### 4.1  MLR trend estimation method

Trends are estimated by fitting a multi-linear regression function to the monthly mean ozone time series considering two piece-

wise linear ramps (PWLT) starting in 1970 and in 1998. Trend profiles are obtained by considering one independent monthly mean time series for each pressure level. The results are given as a difference in DU to the 1970–1980 and of the 2000–2010 means. The explanatory variables represent sources of geophysical variability with known influence on stratospheric ozone, including the quasi-biennial oscillation[1] (QBO) at 30 and 10 hPa, the 10.7 cm solar radio flux describing the 11- year solar cycle[2] (SOL), the El Niño–Southern Oscillation[3](ENSO), the North Atlantic Oscillation[4] (NAO), the Stratospheric Aerosol

Optical Depth[5] (SAOD) and Fourier components representing the seasonal cycle (annual and semi-annual variations). All data points are considered with equal weights, and the uncertainty of the fit parameters is estimated from the regression residuals. Residual autocorrelations are accounted for by applying a Cochrane-Orcutt transformation to the model (Cochrane and Orcutt, 1949).

### 4.2  DLM trend estimation method

Dynamic Linear Modeling allows the determination of a non-linear time-varying trend from a monthly means time series. This is a Bayesian approach regression which fits the data time series for a non-linear time-varying trend, regression coefficients from explanatory variables and seasonal and annual modes, considering their uncertainties and an autoregressive component. The trend is allowed to smoothly vary in time and its degree of non-linearity is inferred from the data, as well as the turnaround period. We use the code by Alsing (2019) which is a Python implementation of the formalism introduced by Laine et al.

(2014) and we refer to these publications for a detailed description of the DLM principles. The used model considers standard regression components, allows a variability of the sinusoidal seasonal modes and includes the autoregressive (AR1) correlation process with variance and correlation coefficient as free parameters in the regression. The same five explanatory variables as in the MLR are used in the trend estimate: QBO at 30 hPa and 10 hPa, SOL, ENSO and SAOD NH values. The estimation

---

[1]from *https://www.geo.fu-berlin.de/met/ag/strat/produkte/qbo/index.html*

[2]from *https://www.iup.uni-bremen.de/UVSAT/Datasets/mgii*

[3]from *http://www.esrl.noaa.gov/psd/enso/mei/*

[4]from *https://climatedataguide.ucar.edu/ climate-data/hurrell-north-atlanticoscillation- nao-index-station-based*

[5]from *https://asdc.larc.nasa.gov/project/GloSSAC/GloSSAC_1.0*

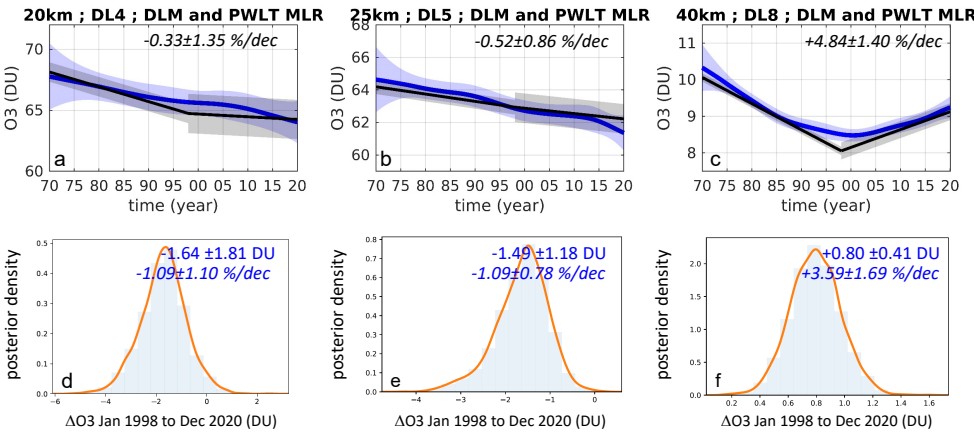

**Figure 6.** (a-c) DLM (in blue) and MLR (in black) trend estimates in %/decade $\pm\ 2\sigma$ of Dobson D051 dataset for 3 DL between 20 and 40 km, the shaded areas show the $2\sigma$ uncertainties, (d-f) the distribution of the DLM trend estimates is given by the kernel density estimation (KDE) for the same 3 DL in the 1998–2020 time range in DU $\pm$ FWHM.

of the posterior uncertainty distribution is performed with the Markov chain Monte Carlo (MCMC) method, and considers the
uncertainties on the regression components, on the seasonal cycle, on the autoregressive correlation and on the non-linearity of the trend. Note that only statistical uncertainties are given in the paper, which allows to determine the significance of the trends. In order to check the agreement of trends derived form different datasets, uncertainties including a term accounting for remaining steps and for inhomogeneities in the dataset (Bernet et al., 2021) should be considered.

### 4.3   Comparison of MLR and DLM trend estimation: case of Dobson D051 dataset

Figure 6 shows the long-term trend estimates from the MCH homogenized Dobson D051 dataset by DLM (in blue with $\pm$2 sigma uncertainty shaded area) and by MLR (PWLT, in black with $\pm$2 sigma uncertainty shaded area) for the same explanatory variables at three altitude levels. The blue shaded areas show the non-constant 2 sigma uncertainties in DU/y estimated by the DLM. By analogy, for the MLR, the grey shaded areas report the uncertainty in DU/y calculated from the constant 2 sigma offset trend uncertainty in DU per decade.

Overall trends are similar but differ over short timescales because of their representation of the nonlinearity of the changes in the data record. The advantage of DLM lies in the estimation of a smoothly varying trend without assuming any shape. The inflection year depends on the method: while the inflection point is fixed by the MLR PWLT (1998 in this case, see Petropavlovskikh et al., 2019), the inflection year is retrieved by the DLM and results in year 2002 for the Dobson D051 dataset

above 28 km. The maximum of the △O3 (ozone difference) between 1998-2020 KDE (kernel density estimation) should be
compared to the linear trend value over the same time period (22 years), while the 95% level of significance, represented by
the fraction of the KDE above/below zero, slightly differs from the MLR uncertainty estimates. In the lower stratosphere, for
DL4 (Fig 6a and d), the post-1998 MLR trends values are -0.33±1.35%/decade. The DLM KDE shows a maximum at -1.64
DU and a full width at half maximum (FWHM) (=2.4 sigma for normal distribution) of 1.81 DU, which means a mean trend of
-1.09±1.10%/decade. MLR estimate is non significantly different from zero at the 95% confidence level while DLM estimate
is negative barely significant at the 95% level. In the middle stratosphere, for DL5 (Fig 6b and e), the post-1998 MLR trends
values are -0.52±0.86%/decade. The DLM KDE shows a maximum at -1.49 DU and a FWHM of 1.18 DU, which means a
mean trend of -1.09±0.78%/decade. MLR estimate is non significantly different from zero at the 95% confidence level while
DLM estimate is significantly negative at the 95% level. In the upper stratosphere, for DL8 (Fig 6c and f), the post-1998
MLR trends values are +4.84±1.40 %/decade and the DLM KDE shows a maximum at +0.80 DU and a FWHM of 0.41 DU,
which means a mean trend of +3.59±1.69%/decade. Both are significantly positive at the 95% confidence level. The estimated
post-1998 MLR trends are in agreement with the vertically resolved trends reported in the litterature (Godin-Beekmann et al.,
2022) and the post-1998 MLR and DLM estimated trends are in agreement within their uncertainties. The lower and middle
stratospheric trends differ in their significance though. In case of high annual variability, a DLM trend estimate in %/decade
may be significant while a MLR trend estimate may be non significant for the same considered period. Note that the given
DLM trend value in %/decade is an average of the percentage change per year. The regressions (resulting trends and their
uncertainties) are influenced by outliers (Bowerman and O'Connell, 1990), but trends estimated by DLM regression change
each year. Hence, outliers influence only a limited portion of the DLM trend time series influencing only the associated trend
uncertainties.

## 5  Long-term trends estimation results

Post-2000 vertically resolved ozone trends for the Arosa/Davos station are estimated by DLM on the MCH and the NOAA
homogenized Dobson D051 Umkehr dataset, on the Brewer B040 Umkehr dataset and on the Aura MLS dataset for overspasses
over the station.

### 5.1  Vertically resolved ozone trends derived from the two homogenized Dobson D051 datasets

Figure 7a and b shows the DLM trend estimates derived from the Dobson D051 record as homogenized by MCH (a) and by
NOAA (b). The trends values are given in percent change per year for each altitude level between 10 and 50 km. Positive and
negative trends are shown with varying intensities of red and blue respectively. The grey lines indicate trend estimates non
significantly different from zero at the 95% confidence level.

The upper stratospheric (DL7-10, 10–1 hPa, 35–50 km) trend estimates are significantly negative between 1965 and 1997
on Figure 7a and before 1997 on Figure 7b. The mean negative trend estimates are -5 %/decade (mean value of the 1965-1997
upper stratospheric trends). Both records show then a transition period until 2003 with non-significant upper stratospheric

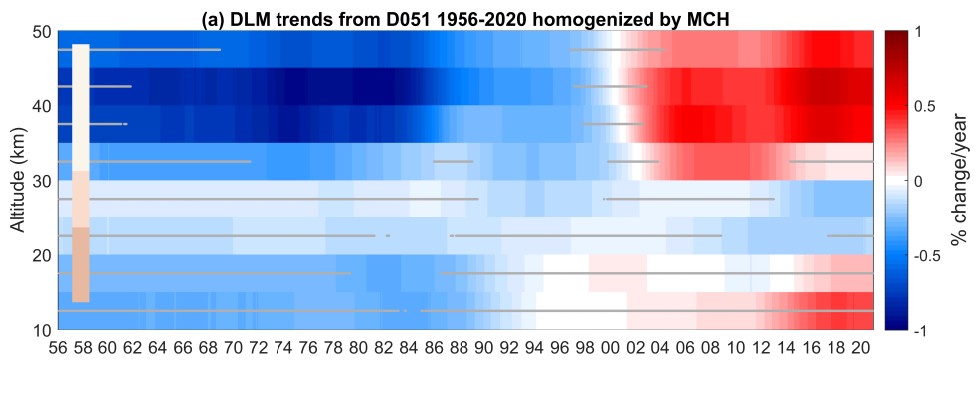

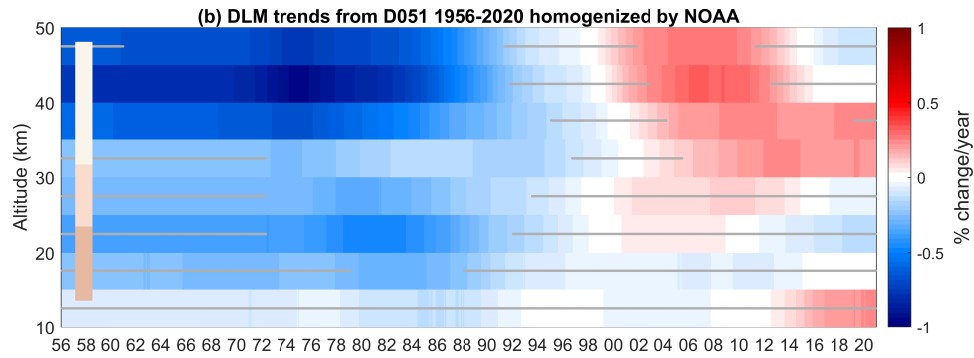

**Figure 7.** DLM trend estimates in %/year of Dobson D051 1956-2020 from (a) MCH homogenized and (b) NOAA homogenized data records. Grey lines indicate trend estimates non significantly different from zero at the 95% confidence level. The orange bars indicate the lower, middle and upper stratospheric ranges.

trend estimates. The post-2003 upper stratospheric trends are significant and positive, up to 2020 for the MCH homogenized Dobson D051 record and until 2013 for the NOAA homogenized Dobson D051 record. The mean positive upper stratospheric trends are 3.6 %/decade on Figure 7a (mean value of the 2003-2013 upper stratospheric trends) and 2.1 %/decade on Figure 7b (mean value of the 2003-2013 upper stratospheric trends). Note that due to the large AK of the Umkehr measurement, the ozone and trend informations in DL8 and DL9 are not independent from each others. In the middle stratosphere (DL5&6, 24–32 km), both homogenized records show a negative trend in DL5, persistent and significantly different from zero at the 95% confidence level since 2012 for the MCH homogenized Dobson D051 data record but slightly positive between 2002 and 2010 and non significantly different from zero at the 95% confidence level for the NOAA homogenized data record. In the lower stratosphere (LS, DL3&4, 14-24 km), the DL3 and DL4 trend estimates are non significantly negative before 1996 but


significantly negative between 2008 and 2018 in DL4 for the MCH homogenized data record and non significantly negative for the NOAA homogenized Dobson D051 record.

Again due to the AKs width of the Umkehr profiles, the ozone content information of DL2 partly overpasses the lower stratosphere as usually defined (see representation of shaded areas in Figure 1b). The same consideration is true for the DL6 in the middle stratosphere. The lower part of the lower stratosphere and the upper part of the middle stratosphere trends may be

aliased by upper tropospheric respectively upper stratospheric information.

## 5.2 Vertically resolved ozone trends derived from the Dobson D051, the Brewer B040 and the Aura MLS datasets.

Post-2000 trend have been estimated on the 3 Dobson and the 3 Brewer MCH Umkehr data records. The trends estimates of one of the Dobsons (D051), one of the Brewers (B040) and Aura MLS are represented in Figure 8 a, b and c in percent change per year for each altitude level between 10 and 50 km.

The post-2000 trends show similar features for the Dobson and Brewer spectrophotometers:

- a positive trend of 0.2 to 0.5 %/year above 35 km, significant for Dobson D051 (and for Dobson D062 and Brewer B156 not shown) but lower and therefore non significantly different from zero at the 95% level of confidence for Brewer B040 and Dobson D101. Despite differences in the trend estimates intensities, an overall picture of a upper stratospheric positive trend after 2002 is shown.

- a persistent negative trend in DL5 of the middle stratosphere and DL4 of the lower stratosphere with different levels of significance depending on the dataset but mostly non significantly different from zero at the 95% confidence level except for Dobson D051.

Significant upper stratospheric positive trends are estimated on the Aura MLS satellite data record (Figure 8c) however non significant since 2013. Signs of negative trends in the lower altitudes are also observed although not significant: DLM trend

estimates are persistently negative in the middle stratosphere and negative in the lower stratosphere since 2012.

## 6 Conclusions

Data records of six collocated spectrophotometers were inter-compared on the raw data level (N values) and on the ozone profile level in order to detect anomalies. The MCH Dobson D051 Umkehr data record has been homogenized on the raw data level by comparison with the collocated Brewer triad data record and with the redundant Dobson data records. In a separate

work, a second homogenization of the same Dobson dataset was performed by NOAA, using comparison with the M2GMI model on the raw data level as well. Both homogenizations result in similar magnitudes of N values corrections relative to the post 2018 values. They differ in the application of a correction for the straylight effect and of a correction of the volcanic eruption periods. By relying on the collocated Brewers datasets, the MCH homogenization accounts for the local variability of the ozone layers content in the 2011-2013 period and results in a smaller correction of the data record for this period. Even if

only slightly different, the homogenizations of the raw data can produce significant differences in ozone profiles and, therefore,

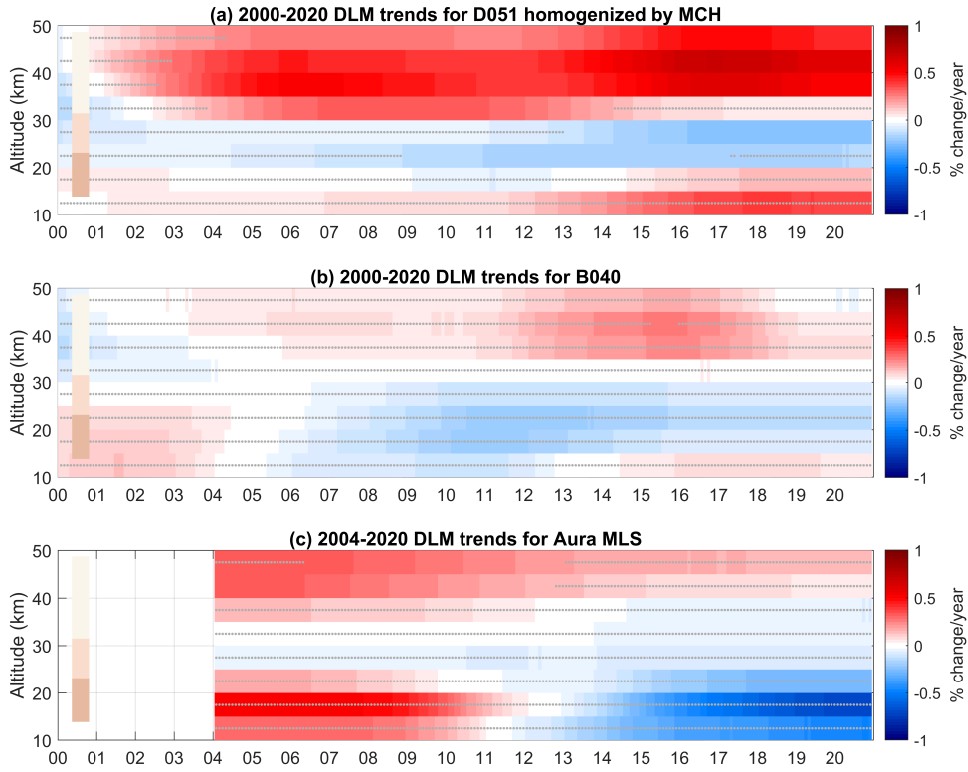

**Figure 8.** Post 2000 DLM trend estimates in %/year from (a) Dobson D051, (b) Brewer B040 and (c) Aura MLS data records.Grey lines indicate trend estimates non significantly different from zero at the 95% confidence level. The orange bars indicate the lower, middle and upper stratospheric ranges.

in the long term trend estimates. The two homogenizations differ in their comparison towards Aura MLS and Brewer B040 on the ozone profiles level in the upper stratosphere, especially for the period 2017-2019.

Trends of the ozone profile time series have been estimated by DLM from the Dobson and the Brewer spectrophotometers datasets. The post-2000 trends show similar features namely a positive trend of 0.2 to 0.5 %/year above 35 km in the upper stratosphere, significant for Dobson D051 but lower and therefore non significantly different from zero at the 95% level of confidence for Brewer B040, and a persistent negative trend in DL5 of the middle stratosphere with different levels of significance depending on the dataset. The DLM trend estimates from Dobson D051 show a significant persistent negative trend in DL5 and supports also the mention of a persistent negative trend in the NH lower stratosphere (in DL4) when measured by ground-based instrument, considering, however, that the trends estimates in the upper part of the middle stratosphere and in the lower part of the lower stratosphere are aliased by the large AKs of the Umkehr profiles.

DLM trend estimates derived from Aura MLS show similar features in the upper stratosphere and the middle stratosphere as estimates from the ground-based Dobson and Brewer spectrophotometers. However, a transition from non significant positive to non significant negative trends in the lower stratosphere remains unexplained.

While significant positive trends have been estimated in the upper stratosphere since 2004 from the MCH homogenized Dobson D051 dataset, the trend estimates from the NOAA homogenized data record appear to show a transition from significant positive to non-significant negative/zero values above 40 km in 2016. Further investigation will be needed to confirm this transition and exclude 2017 as a problematic period in the NOAA homogenization.

Both homogenization approaches considered in this study are relevant and significantly improve the Dobson D051 data record. However, inconsistencies in the level of significance of the Dobson D051 trend estimates are noticed and should be attributed to the remaining differences left by the homogenizations in the data records.

*Data availability.* The as measured Dobson D051 dataset is available at WOUDC. The NOAA homogenized Dobson D051 dataset is available at https://gml.noaa.gov/aftp/data/ozwv/Dobson/AC4/Umkehr/Optimized/Daily/ARO/. The MCH homogenized Dobson D051 and the Brewer B040 datasets are available at https://doi.org/10.5281/zenodo.7185409. The MLS ozone dataset is available from the NASA Goddard Space Flight Center Earth Sciences Data and Information Services Center (GES DISC) at http://disc.sci.gsfc.nasa.gov/Aura/data-holdings/MLS/index.shtml.

*Author contributions.* E.M.B. is responsible for the Umkehr ozone measurements with the Arosa/Davos Dobson and Brewer spectrophotometers, performed the data analysis and prepared the manuscript. A.H. and R.S. contributed to the interpretation of the results. A.J. performed the 2011-2013 homogenization and the first DLM trend derivation. H.S. is responsible for the Arosa/Davos Dobson and Brewer spectrophotometers. I.P. and K.M. performed the NOAA homogenization of D051 and contributed to the interpretation of the results. M.S. implemented the Umkehr Brewer retrieval algorithm. L.F. is responsible for the Aura MLS measurements. All co-authors contributed to the preparation of the manuscript.

*Competing interests.* The authors have no competing interests.

*Acknowledgements.* This work has been funded by MeteoSwiss within the Swiss Global Atmospheric Watch program of the World Meteorological Organization. Work performed at NOAA has been supported by the Climate Program Office (grant no. NA19OAR4310169). Work performed at the Jet Propulsion Laboratory, California Institute of Technology, was performed under contract with the National Aeronautics and Space Administration.

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
