# Peer review of "Dynamic Linear Modeling estimates of long-term ozone trends from homogenized Dobson Umkehr profiles at Arosa/Davos, Switzerland"

_Atmospheric Chemistry and Physics, 2022_

## Referee Comment (RC1)

Review of „DLM estimates of long-term Ozone trends from Dobson and Brewer Umkehr profiles" by E. Maillard Barras et al.

The paper focuses on post-2000 ozone profile trend estimates using the Dynamical Linear Modeling (DLM) approach. The method is applied to Umkehr time series obtained from six collocated Dobson and Brewer spectrophotometers based in Arosa/Davos, Switzerland. One of the Dobson instruments (D051) has been homogenized based on two different approaches (the MCH homogenization based on comparisons with the 5 collocated Dobson and Brewer instruments and the NOAA homogenization developed by Petropavlovskikh et al., 2022) before trend estimation. In addition, the DLM approach is applied to the Aura/MLS satellite data record.

The post-2000 trends show similar features (e.g. an increase in ozone in the upper stratosphere) for most altitudes, although with inconsistencies in the level of significance. The authors attribute the differences to remaining inhomogeneities in the ground-based data records. For the lower stratosphere, no clear picture was found.

The paper fits well into the scope of ACP and I recommend publication after having addressed the comments (mostly minor) below.

**Specific comments:**

P1, L6: I would suggest to mention here the time period which is covered by D051.

P1, L14: What is meant with globally here?

P1, L19: "Moreover, a persistent negative trend is estimated in the middle and lower stratosphere with different…" → In the lower stratosphere the trend obtained from D051 is positive though non significant (Fig. 8a).

P2, L45: Do you mean the sensitivity with respect to the length of the fitting period and with respect to the start/end dates?

P3, L88: Could you please briefly mention, why exactly this specific instrument D051 was selected for homogenization?

P4, Sec.2.1: A table which indicates the periods during which the individual instruments at Arosa/Davos are measuring (and how often per week/month) could be helpful.

P5, L132 When the SZA is increasing from 60° to 90° the intensities should decrease.

P6, Figure 1: What is the difference between these two examples (blue/black curve)? The total ozone column, other atmospheric conditions,…?

P7, Subsection 2.3: I would suggest to remove the entire subsection (see the comment P18, LL386-390 for an explanation).

P8, Section 3: Think about subdividing Section 3 into three subsections (e.g., 3.1 MCH homogenization, 3.2 NOAA homogenization, and 3.3 Comparison of both homogenizations)

P8, L196: Could you provide a reference for this 2008 homogenization?

P8, L211: You state that the black frames in Fig. 2 indicate periods that can be attributed to technical issues, but Table 1 indicates that there was no technical issue in 2010; could you please clarify?

P9, Figure 2: In 2014 the comparison of D051 with the three Brewer instruments indicates a strong positive bias from 0-30km and a strong negative bias from 30-50km. Did you investigate this period in more detail?

P10, Table 1: What is the meaning of "RtoN" table? Last row: what is the start date for the homogenization "before 2018/05/01"?

P11, L261: I would suggest to write "The MCH homogenization approach" instead of "Our approach".

P12, Fig. 4: Is the MCH correction from 1956 to 1988 applied to D051 or to D015? I would suggest to briefly explain/mention the origin of the correction during that early period, because it was not discussed in the text.

P13, L293: A similar feature is seen in 2014; did you investigate this period in more detail?

P13, L298: Please add the panel of Fig. 5 you are referring to here.

P14, L306: The results, that you show, are for an inflection point of 1998.

P15, Fig.6: Please mention in the caption that the black curve belongs to PWLT and the blue to DLM.

P15, L328: Which D051 time series is used here? Uncorrected, MCH homogenized, or NOAA homogenized?

P15, L340: Maybe it is sufficient to define statistical significance (95% confidence level) only once, and then you don't have to mention "at the 95% confidence level" at every single occurrence.

P16, LL348-350: Do you mean here that MLR is more significantly impacted by outliers or boundary values than DLM? Could you provide a reference?

P16, Fig. 7 caption: Please add the info which Dobson Layers are shown. Moreover, I would suggest to add horizontal lines in the plots which delineate the different stratospheric sectors LS, MS, UpS.

P17, Fig. 8 a+b: Do the DLM trend results for D051 (MCH homogenized) and Brewer B040 change when you use the period 2004-2020? Just to make sure that it is consistent with the trend period used for MLS (panel c).

P17, LL366-368: "The lower stratospheric (LS, DL3&4, 14-24 km) trend estimates are non significantly negative before 1996 but significantly negative between 2008 and 2018 for the MCH homogenized data record and non significantly negative for the NOAA homogenized Dobson D051 record." → For DL 3 and DL 4 in Fig. 7a I cannot see the significant negative trend for MCH D051 between 2008 and 2018. Could you please doublecheck?

P17, L374: Please add that MLS trends are shown in panel c.

P18, L382: Please add that trends are significant only in DL6.

P18, LL386-390: The comparison with the trends derived from the Boulder and OHP time series is quite limited. In my view this paragraph could be either entirely deleted (since the main focus of this paper is the Arosa record) or a much more detailed comparison (including a plot showing the trends)

should be provided. However, for the comparison of trends from various locations the latitudinal and longitudinal variability of the altitude dependent trends should be kept in mind (Sofieva et al., 2021).

P18, LL402-403: To which plot do you refer to here? The agreement between the NOAA homogenized D051 and Aura MLS (Fig. 5) is quite good.

**Technical issues:**

\* Check that all acronyms are defined, e.g., P2: M2GMI, MERRA2, SCIAMACHY, OMPS,…

\* Sometimes the instrument numbers are written including the "B" or "D" (e.g. D051) and sometimes without that letter. I would suggest to use the notation including the letter consistently throughout the manuscript.

P2, L24: "(MP1, 1987)" → key "MP1" not found in list of references

P2, L29: "discrepencies" → "discrepancies"

P2, L37: "discrepencies" → "discrepancies"

P2, L43: remove blank before comma

P2, L50: "applied it on" →  "applied it to"

P3, L56: "satellites data records" → "satellite data records"

P3, L63: "record" -> "records"

P4, L110: remove blank after "D051"

P5, L152: define acronym "Mk"

P6, Fig. 1 caption: "profiles in DU in functions of" → "profiles in DU as a function of"

P6, L153: "wavelenths" → "wavelengths"

P7, L187: remove parenthesis from "Waters et al., 2006"

P8, L204: do you mean "not flagged" here?

P9, Fig. 2 caption: "time serie are" → "time series are"

P11, L258: "homogenization remove" → "homogenization removes"

P11, L272: "both homogenizations differs" → "both homogenizations differ"

P12, L279: "both corrections of the N values looks" → "both corrections of the N values look"

P12, L282: "both homogenization" → "both homogenizations"

P14, L324: "30hPA" → "30hPa"

P16, L362: "informations" → "information"

P17, Fig. 8 caption: "post" → "Post"

P19, L421: "homog" → "homogenized"

P21, L504: "serie" → "series"

P21, L509: DOI missing

P22, L510: add blank after "S.M."

P22, L537: "Mcclure" → "McClure"

P22, L542: "Deluisi" → "DeLuisi"

P24, L605: journal missing

**References:**

Sofieva, V. F., Szeląg, M., Tamminen, J., Kyrölä, E., Degenstein, D., Roth, C., Zawada, D., Rozanov, A., Arosio, C., Burrows, J. P., Weber, M., Laeng, A., Stiller, G. P., von Clarmann, T., Froidevaux, L., Livesey, N., van Roozendael, M., and Retscher, C.: Measurement report: regional trends of stratospheric ozone evaluated using the MErged GRIdded Dataset of Ozone Profiles (MEGRIDOP), Atmos. Chem. Phys., 21, 6707–6720, https://doi.org/10.5194/acp-21-6707-2021, 2021.

---

## Author Comment (AC1)

**Response to referee #1**

DLM estimates of long-term Ozone trends from Dobson and Brewer Umkehr profiles.
Eliane Maillard Barras et al., Atmos. Chem. Phys. Discuss., https://doi.org/10.5194/acp-2022-344.

Dear Referees,
Dear Editor Gaby Stiller,

We would like to thank the referees for the detailed review of the manuscript and for their constructive and helpful comments and suggestions. We have taken the remarks into account and we are presenting the detailed answers in the following. We attach a revised version of the manuscript with marked changes.
We hope that we have satisfactorily addressed the suggestions and remarks.
The referee's comments are given in italic, our responses are given in blue, and the corresponding changes in the manuscript in grey.

Best regards,
Eliane Maillard Barras (on behalf of all co-authors)

*The paper focuses on post-2000 ozone profile trend estimates using the Dynamical Linear Modeling (DLM) approach. The method is applied to Umkehr time series obtained from six collocated Dobson and Brewer spectrophotometers based in Arosa/Davos, Switzerland. One of the Dobson instruments (D051) has been homogenized based on two different approaches (the MCH homogenization based on comparisons with the 5 collocated Dobson and Brewer instruments and the NOAA homogenization developed by Petropavlovskikh et al., 2022) before trend estimation. In addition, the DLM approach is applied to the Aura/MLS satellite data record.*
*The post-2000 trends show similar features (e.g. an increase in ozone in the upper stratosphere) for most altitudes, although with inconsistencies in the level of significance. The authors attribute the differences to remaining inhomogeneities in the ground-based data records. For the lower stratosphere, no clear picture was found.*
*The paper fits well into the scope of ACP and I recommend publication after having addressed the comments (mostly minor) below.*

***Specific comments:***
*P1, L6: I would suggest to mention here the time period which is covered by D051.*
The time period has been added.

In this study, the worldwide longest Umkehr dataset (1956-2020) is carefully homogenized using collocated and simultaneous Dobson and Brewer measurements.

*P1, L14: What is meant with globally here?*
Globally is too general here, we removed it.

The two homogenized data records show common correction periods, except for the 2017-2018 period, and produce corrections similar in magnitude.

*P1, L19: "Moreover, a persistent negative trend is estimated in the middle and lower stratosphere with different…" → In the lower stratosphere the trend obtained from D051 is positive though non significant (Fig. 8a).*

The lower stratosphere is composed of DL3 and DL4 (see Fig1). The trend in DL4 is negative, significant between 2009 and 2017 and non significant after. The DL3 trend is slightly positive and non significant. As it is difficult to draw a conclusion for the whole lower stratosphere on that basis, we now draw conclusions at the Dobson layer level.

In the lower stratosphere, the trend is negative at 20km with different levels of significance depending on the period and on the dataset.

*P2, L45: Do you mean the sensitivity with respect to the length of the fitting period and with respect to the start/end dates?*

We refer here to the sensitivity to the start/end dates as Bernet et al. 2019 report on the trend differences with varying starting years and Dietmüller at al. 2021 report on influence of the year to year variability on the trend values and their significance. We modified the sentence accordingly.

The sensitivity of the post-2000 trend magnitude to the start and end years has been extensively discussed (Petropavlovskikh et al., 2019; Bernet et al., 2019; Dietmueller et al.,2021).

*P3, L88: Could you please briefly mention, why exactly this specific instrument D051 was selected for homogenization?*

Yes, you are right. This was not clear. We amended the text accordingly.

In Arosa/Davos, the Dobson D051 is the station's primary instrument for continuous Umkehr profile time serie. It was dedicated exclusively to Umkehr measurement from 1988 until February 2013, when total ozone measurement was added to the schedule. The number of observations dedicated to Umkehr was not impacted and the number of retrieved Dobson D051 Umkehr profiles was kept to two profiles per day up to now. This frequency in observations allows the computation of statistically reliable monthly means for trend estimations. However, the instrument operations recently suffered from anomalies following technical interventions. Therefore, a complete homogenization of the Dobson D051 Umkehr data record has been performed and is described in this paper.

*P4, Sec.2.1: A table which indicates the periods during which the individual instruments at Arosa/Davos are measuring (and how often per week/month) could be helpful.*

As suggested by both reviewers, we added a table in the section 2.1.

| Instrument | | Time range | Time resolution |
|---|---|---|---|
| Dobson | D015 | 1956-1988 | 2 profiles/day |
| | D051 | 1988-now | 2 profiles/day |
| | D062 | 1998-now | 4-6 profiles/month |
| | D101 | 1988-now | 4-6 profiles/month |
| Brewer | B040 | 1988-now | 2 profiles/day |
| | B072 | 2005-now | 2 profiles/day |
| | B156 | 2005-now | 2 profiles/day |

**Table 1.** Time ranges and time resolutions of the Dobson and Brewer Umkehr measurements at the Arosa/Davos station.

*P5, L132 When the SZA is increasing from 60° to 90° the intensities should decrease.*
Yes, correct. We amended the description.

As the SZA is increasing from 60° to 90°, the scattering height is increasing, and the two intensities decrease because of increased absorption and scattering by ozone and air molecules.

*P6, Figure 1: What is the difference between these two examples (blue/black curve)? The total ozone column, other atmospheric conditions,…?*
These are the morning and afternoon measurements of a random day with slight differences in both TCO and atmospheric conditions. The purpose is to show that an apparent small difference in the N curves can lead to a significant difference in the ozone profiles. This is now mentioned in the caption.

(a) Morning (in black) and afternoon (in blue) N curves at 12 nominal SZAs and (b) their corresponding retrieved ozone profiles in DU as a function of altitude in km and pressure level in hPa. Total column ozone and atmospheric conditions slightly differs between the morning and the afternoon. Altitude ranges of the 10 Dobson layers (DL) are shown in (b). Lower, middle and upper stratospheric ranges are displayed in orange shadings.

*P7, Subsection 2.3: I would suggest to remove the entire subsection (see the comment P18, LL386-390 for an explanation).*
Agreed. Done. Same for P18 L386-390.

*P8, Section 3: Think about subdividing Section 3 into three subsections (e.g., 3.1 MCH homogenization, 3.2 NOAA homogenization, and 3.3 Comparison of both homogenizations)*
Agreed. Done.

*P8, L196: Could you provide a reference for this 2008 homogenization?*
As the 1988 D015 to D051 homogenization has been reprocessed for this study, the 2008 homogenization is finally only a reprocessing of the N values with adapted shaft encoder positioning. This was not published but reported as an internal report. We remove any mention of the 2008 "homogenization" as it should be considered only as a reprocessing and we discuss in more details the correction of the D015 to D051 transition.

The Arosa/Davos Umkehr time series is composed of Dobson D015 measurements from 1956 to 1988 and Dobson D051 since then. The quality of the homogenization of the Dobson D015 to Dobson D051 transition has been ensured by one year of parallel measurements (1988) allowing an adaptation of the D015 N values to the D051 N values. For each SZA, the 1988 mean difference between the D051 and the D015 N values has been added to the D015 values. The 1956-1987 ozone profiles have then been retrieved from the Dobson D015 corrected N values. No statistical correction has been performed on the D015 ozone dataset.
We report here about the complete homogenization of the 1988-2020 Umkehr Dobson D051 time series by comparison to the datasets of the five collocated instruments (two Dobson and three Brewer spectrophotometers) on the N value level.

*P8, L211: You state that the black frames in Fig. 2 indicate periods that can be attributed to technical issues, but Table 1 indicates that there was no technical issue in 2010; could you please clarify?*
Yes, correct, this is in contradiction. The black frames indicate the anomalies. Most of them can be attributed to technical issues but not in 2010. Text was modified.

If we focus on the post-2000 period, where several collocated and redondant measurements are available, systematic anomalies of the Dobson D051 are noticed (periods in black frames in Figure 2).

*P9, Figure 2: In 2014 the comparison of D051 with the three Brewer instruments indicates a strong positive bias from 0-30km and a strong negative bias from 30-50km. Did you investigate this period in more detail?*
This period should have been removed. This is an error. The figure has been corrected now. The 2014 anomaly should be considered with caution (P8 L219-220) because of the very reduced number of measurements at that period (many data are missing or have to be flagged because of technical issues during the refurbishment period and during this technical staff transition period). It is therefore very difficult to investigate this period. This is now mentioned in the text.

[Figure]

**Figure 2.** Monthly mean time series of the ozone profiles relative differences for each of the 5 spectrophotometers with respect to D051. The time series are deseasonalized and smoothed by a 6 months moving average.

The comparison of Dobson D051 with the collocated Dobsons around 2014 and after 2018 are to be taken with caution due to the very limited number of measurements of Dobson D051 in 2014 and of Dobson D062 and Dobson D101 during these periods. Around 2014 (technical and staff transition period), many data are missing or have to be flagged because of roof opening issues. After 2018, the Umkehr measurement by Dobson D062 and Dobson D101 have been drastically reduced as priority has

been given to total ozone measurements.

*P10, Table 1: What is the meaning of "RtoN" table?*
The "RtoN" table is the table mentioned on P5 L138. We added here the definition of the acronym "RtoN".

The logarithm of the ratio of the two wavelengths intensities (R values) is converted to radiance using calibration tables (RtoN table) and reported as N values…

*Last row: what is the start date for the homogenization "before 2018/05/01"?*

The correction offset is calculated using the 2016.05.01-2018.05.01 (and 2018/05/01-2020/05/01) period and applied to the 1956-2018/04/30 dataset.

*P11, L261: I would suggest to write "The MCH homogenization approach" instead of "Our approach".*
Modified as suggested.

The MCH homogenization approach is different in that the homogenization process aims to remove artificial steps in the Dobson D051 Umkehr profiles record while maintaining the constant offset between the datasets, …

*P12, Fig. 4: Is the MCH correction from 1956 to 1988 applied to D051 or to D015? I would suggest to briefly explain/mention the origin of the correction during that early period, because it was not discussed in the text.*
The MCH correction from 1956 to 1988 is applied to D015. We discuss in more details the correction of the D015 to D051 transition. See response to "P8 L196" comment.

*P13, L293: A similar feature is seen in 2014; did you investigate this period in more detail?*
No, we did not. See response to "P9, Figure 2" comment.

*P13, L298: Please add the panel of Fig. 5 you are referring to here.*
Reference to Fig.5 (c) has been added.

Due to the occurence of an anomaly in 2018, which is particularly visible in DL8 for all datasets (Fig. 5 c), the last correction applied to the dataset by the NOAA and the MCH homogenizations differ.

*P14, L306: The results, that you show, are for an inflection point of 1998.*
Yes correct, this is a residual from a previous text version. "2000" has been replaced by "1998" in the 4.1 and 4.2 sections. Trend values are correct only the text had not been adapted.

*P15, Fig.6: Please mention in the caption that the black curve belongs to PWLT and the blue to DLM.*
Ok, we amended the text.

(a-c) DLM (in blue) and MLR (in black) trend estimates in %/decade ± 2σ of Dobson D051 dataset for 3 DL between 20 and 40 km

*P15, L328: Which D051 time series is used here? Uncorrected, MCH homogenized, or NOAA homogenized?*

The MCH homogenized time series is used. Mention is done now.

Figure 6 shows the long-term trend estimates from the MCH homogenized Dobson D051 dataset by DLM…

*P15, L340: Maybe it is sufficient to define statistical significance (95% confidence level) only once, and then you don't have to mention "at the 95% confidence level" at every single occurrence.*
We prefer to mention it each time. It is heavy but it appears 7 times only and so the text does not allow for misunderstanding.

*P16, LL348-350: Do you mean here that MLR is more significantly impacted by outliers or boundary values than DLM? Could you provide a reference?*
This is not exactly what we mean. Actually, regressions (resulting trends and their uncertainties) are influenced by outliers. But, trends estimated by DLM regression change each year. In case of influential outliers, only the particular year trend value and its uncertainty is influenced by the outliers. This has been rephrased.

Regressions (resulting trends and their uncertainties) are influenced by outliers (Bowerman and O'Connell, 1990). But, trends estimated by DLM regression change each year. Hence, outliers influence only a limited portion of the DLM trend time series.

*P16, Fig. 7 caption: Please add the info which Dobson Layers are shown. Moreover, I would suggest to add horizontal lines in the plots which delineate the different stratospheric sectors LS, MS, UpS.*
We added now vertical color bars with the same color scheme as in Figure 1 in order to delineate the LS, MS and UpS altitude ranges. We amend the caption accordingly.

[Figure]

Figure 7. DLM trend estimates in %/year of Dobson D051 1956-2020 from (a) MCH homogenized and (b) NOAA homogenized data records. Grey lines indicate trend estimates non significantly different from

zero at the 95% confidence level. The orange bars indicate the lower, middle and upper stratospheric ranges.

*P17, Fig. 8 a+b: Do the DLM trend results for D051 (MCH homogenized) and Brewer B040 change when you use the period 2004-2020? Just to make sure that it is consistent with the trend period used for MLS (panel c).*

MCH homogenized D051 and B040 DLM trends for the period 2004-2020 are shown below. The trend values are similar to the DLM trends for the period 2000 to 2020, with slight differences in their significance though. The choice of the period does not explain the lack of consistency with the MLS DLM trend results. As the DLM regression results in a trend variation by year, it is not drastically influenced by the starting year. Note that the color scale range is small and therefore a small difference between non significant positive or negative trends can visually appear as a big difference.

[Figure]

Post 2004 DLM trend estimates in %/year from Dobson D051 and Brewer B040

*P17, LL366-368: "The lower stratospheric (LS, DL3&4, 14-24 km) trend estimates are non significantly negative before 1996 but significantly negative between 2008 and 2018 for the MCH homogenized data record and non significantly negative for the NOAA homogenized Dobson D051 record." → For DL 3 and DL 4 in Fig. 7a I cannot see the significant negative trend for MCH D051 between 2008 and 2018. Could you please doublecheck?*

Yes, you are right. See response to "P1 L19" comment. DL3 and DL4 trend differences make it difficult to draw a conclusion for the whole lower stratosphere, we describe trends at the Dobson layer level now.

In the lower stratosphere (DL3&4, 14-24 km), the DL3 and DL4 trend estimates are non significantly negative before 1996 but significantly negative between 2008 and 2018 in DL4 for the MCH homogenized data record and non significantly negative for the NOAA homogenized Dobson D051 record.

*P17, L374: Please add that MLS trends are shown in panel c.*
MLS trends are now mentioned to be shown in Fig 8 c.

The trends estimates of one of the Dobsons (D051), one of the Brewers (B040) and Aura MLS are represented in Figure 8 a, b and c in percent change per year for each altitude level between 10 and 50 km.

*P18, L382: Please add that trends are significant only in DL6.*
We refer here to the negative trends in MCH homogenized D051 DL4 and DL5. In the post 2000 time range, there are significantly negative for the 2008-2018 (DL4) and since 2014 (DL5).
- a persistent negative trend in DL5 of the middle stratosphere and DL4 of the lower stratosphere with different levels of significance depending…

*P18, LL386-390: The comparison with the trends derived from the Boulder and OHP time series is quite limited. In my view this paragraph could be either entirely deleted (since the main focus of this paper is the Arosa record) or a much more detailed comparison (including a plot showing the trends) should be provided. However, for the comparison of trends from various locations the latitudinal and longitudinal variability of the altitude dependent trends should be kept in mind (Sofieva et al., 2021).*

We decided to remove this paragraph and the P7 Subsection 2.3 as we want to keep the focus of the paper on the Arosa data record.

*P18, LL402-403: To which plot do you refer to here? The agreement between the NOAA homogenized D051 and Aura MLS (Fig. 5) is quite good.*
We are referring to Fig5b, where the difference of MCH homogenized D051 and NOAA homogenized towards MLS disagree especially between 2017 and 2019: the difference between the blue curve and the red curve is not constant throughout the 2004-2021 period and is smaller during the 2017-2019 period. We rephrased it for clarity.

The two homogenizations differ in their comparison towards MLS and Brewer B040 on the ozone profiles level in the upper stratosphere, especially for the period 2017-2019.

**Technical issues:**
*\* Check that all acronyms are defined, e.g., P2: M2GMI, MERRA2, SCIAMACHY, OMPS,…*
All acronyms are now defined.
*\* Sometimes the instrument numbers are written including the "B" or "D" (e.g. D051) and sometimes without that letter. I would suggest to use the notation including the letter consistently throughout the manuscript.*
The notation including the letter is now used consistently.
*P2, L24: "(MP1, 1987)" → key "MP1" not found in list of references*
The reference label has been adapted.
*P2, L29: "discrepencies" → "discrepancies"*
The typo has been corrected.
*P2, L37: "discrepencies" → "discrepancies"*
The typo has been corrected.
*P2, L43: remove blank before comma*
The typo has been corrected.
*P2, L50: "applied it on" → "applied it to"*
This has been corrected as suggested.
*P3, L56: "satellites data records" → "satellite data records"*
This has been corrected

*P3, L63: "record" -> "records"*
The typo has been corrected.
*P4, L110: remove blank after "D051"*
The typo has been corrected.
*P5, L152: define acronym "Mk"*
"Mk" has been replaced by "Mark"
*P6, Fig. 1 caption: "profiles in DU in functions of" → "profiles in DU as a function of"*
This has been corrected as suggested.
*P6, L153: "wavelenths" → "wavelengths"*
The typo has been corrected.
*P7, L187: remove parenthesis from "Waters et al., 2006"*
This has been corrected.
*P8, L204: do you mean "not flagged" here?*
Yes, these data are removed. This has been corrected.
*P9, Fig. 2 caption: "time serie are" → "time series are"*
The typo has been corrected.
*P11, L258: "homogenization remove" → "homogenization removes"*
The typo has been corrected.
*P11, L272: "both homogenizations differs" → "both homogenizations differ"*
The typo has been corrected.
*P12, L279: "both corrections of the N values looks" → "both corrections of the N values look"*
The typo has been corrected.
*P12, L282: "both homogenization" → "both homogenizations"*
The typo has been corrected.
*P14, L324: "30hPA" → "30hPa"*
The typo has been corrected.
*P16, L362: "informations" → "information"*
This is correct.
*P17, Fig. 8 caption: "post" → "Post"*
The typo has been corrected.
*P19, L421: "homog" → "homogenized"*
This has been corrected as suggested.
*P21, L504: "serie" → "series"*
The typo has been corrected.
*P21, L509: DOI missing*
DOI has been added.
*P22, L510: add blank after "S.M."*
The typo has been corrected.
*P22, L537: "Mcclure" → "McClure"*
The typo has been corrected.
*P22, L542: "Deluisi" → "DeLuisi"*
The typo has been corrected throughout the References section.
*P24, L605: journal missing*
Journal has been added

---

## Author Comment (AC2)

**Response to reviewer #2**

DLM estimates of long-term Ozone trends from Dobson and Brewer Umkehr profiles.
Eliane Maillard Barras et al., Atmos. Chem. Phys. Discuss., https://doi.org/10.5194/acp-2022-344.

Dear Referees,
Dear Editor Gaby Stiller,

We would like to thank the referees for the detailed review of the manuscript and for their constructive and helpful comments and suggestions. We have taken the remarks into account and we are presenting the detailed answers in the following. We attach a revised version of the manuscript with marked changes.
We hope that we have satisfactorily addressed the suggestions and remarks.
The referee's comments are given in italics, our responses are given in blue, and the corresponding changes in the manuscript in grey.

Best regards,
Eliane Maillard Barras (on behalf of all co-authors)

***General Comments***

*The primary subject of the submitted manuscript is the long-term trend in vertically resolved stratospheric ozone based on the measurements of Dobson and Brewer spectrophotometers operated at Arosa and Davos using the Umkehr technique.*

*Overall, this is an important topic and a very valuable dataset. Further, the work is essentially sound and certainly well within the scope of ACP.*

*I recommend publication, but only after a significant revision to improve the clarity.*

*Unfortunately, in its present form, I have to say I found the manuscript very difficult to follow. In numerous places the discussion went straight into details without sufficient introduction, terms were used without explanation (including acronyms) or the reader was assumed to be aware of facts that hadn't been presented yet.*

We have now added introductions before going into details in all sections and paragraphs listed under the specific comments and listed as examples in the general comments.

*Even the overall scope of the work is unclear.*

We changed the title, rewrote the abstract (see response to "Lines 7-11 comment") and reorganize the sections in order to make the scope of the paper clearer.

Dynamic Linear Modeling estimates of long-term ozone trends from homogenized Dobson Umkehr profiles at Arosa/Davos, Switzerland.

**Contents**

*As a prime example, the place of the alternative NOAA homogenization remains mysterious to me even after several readings. In lines 260-262 the text states "The NOAA homogenization has been developed … Our approach is different …". This can only mean the NOAA work is not part of the current work, but is being presented as an alternative for comparison. The MCH homogenization is described in much greater detail. However, I note two co-authors are included for performing the NOAA homogenization, according to the stated author contributions.*

The presented paper does not include the NOAA work and we modified the manuscript in order to make it clearer. The description of the NOAA homogenization has been published this year in Petropavlovskikh, I., Miyagawa, K., Mcclure-Beegle, A., Johnson, B., Wild, J., Strahan, S., Wargan, K., Querel, R., Flynn, L., Beach, E., G.,A., and Godin-Beekmann, S.: Optimized Umkehr profile algorithm for ozone trend analyses, Atmospheric Measurement Techniques, 15, https://doi.org/https://doi.org/10.5194/amt-15-1849-2022, 2022. Dobson Umkehr ozone profiles from four NOAA ozone network stations (Boulder, the Haute-Provence Observatory (OHP), the Mauna Loa Observatory (MLO), and Lauder) have been homogenized and are presented in Petropavlovskikh et al. 2022. The same NOAA homogenization technique has been applied to the D051 dataset but is not included in Petropavlovskikh et al., 2022. The NOAA homogenized D051 dataset is however described in Garane, K., Koukouli, M., Fragkos, K., Miyagawa, K., Fountoukidis, P., Petropavlovskikh, I., Balis, D., and Bais, A.: Umkehr Ozone Profile Analysis and Satellite Validation, ESA project WP-2190, https://zenodo.org/record/5584472, 2022 and we refer to this publication (P4 L91) without describing the NOAA homogenization process in much details. We feel that comparing our results to the results from these two publications is important, interesting and increases the confidence in both homogenization methods. Two co-authors of Petropavlovskikh et al. 2022 are included in our

publication for contributing to the description and analyses of the NOAA homogenization of the D051 dataset.

*Even more confusingly, section 2.3 gives details about measurements at Boulder and OHP which seem of almost no relevance. (Incidentally "OHP" is never explained or located on the globe).*
We removed this section and the short discussion of the results P18 L386-390.

*It seems to me the core of the work is the post-2000 trend using DLM based on Dobson 051 using the MeteoSwiss homogenization, but the manuscript also at times covers the pre-2000 trend as well as post-2000, the MLR trend as well as DLM, the Brewer and AURA- MLS trends as well as the Dobson, and the NOAA homogenization as well as the MCH. If such a broad scope is going to be covered then the authors need to provide a lot of assistance to the reader to make sure the right message is being communicated.*

We modified the title, the abstract and part of the introduction in order to emphasize the homogenization work rather than the assessment of trends and to provide the reader a thread that we hope is easier to follow.

*As an example, from my understanding of your description, the NOAA homogenization is designed to minimise the offset with AURA-MLS (line 258). The discussion of Figure 5 implies AURA-MLS is being used as a reference to compare the quality of the four datasets shown, which is inconsistent with the first approach. Then Figure 8 compares the trend determined from Dobson 051, Brewer 040 and AURA-MLS, as if these are independent datasets. These three discussions will seem to be mutually contradictory to the reader without better explanation.*

The NOAA homogenization is not designed to minimize the offset with AURA MLS but the iterative optimization of the stray light correction for the D051 observation results in a reduced bias towards Aura MLS . The generic stray light correction (Petropavlovskikh et al, 2011 and 2022) was developed based on the generic Dobson optical design (i.e. 10e-5 rejection of stray light) and can underestimate/overestimate the stray light present in a specific instrument. By applying the additional correction, the bias between Umkehr and other ozone records in the upper stratosphere is reduced. To verify if the correction is adequate for D051, other ozone records (i.e. SBUV, ozonesonde, SAGE II/III and MLS) are used in the iterative process of the stray light correction. However, the adjustment is not performed to match with the MLS specifically. We modified the description of the NOAA homogenization in that sense.

An iterative modification of the N value correction is further performed for optimization of the stray light correction, adding a constant offset correction to the Umkehr dataset. This results in a reduced bias to other ozone records in the upper stratosphere but, as a constant offset, does not have any impact on the trends.

We say in the discussion of Fig 5. (section 3.3) that the small mean bias between NOAA homogenized D051 dataset and Aura MLS is a result of the NOAA optimization procedure. It is not the amplitude of the offset but its potential variation (the offset should be constant) which is investigated here. We add a precision in that sense in the discussion of Figure 5.

The small mean bias is a result of the NOAA optimization of the stray light correction. Therefore, it is not the magnitude of the bias between the homogenized dataset and Aura MLS but its variation (the bias should be constant) which should be considered here.

Therefore, we believe the discussions of Fig 5 not to be in contradiction with the NOAA homogenization principles. The Dobson D051 homogenized by MCH and the Brewer B040 are independent from Aura

MLS as is the NOAA homogenized Dobson D051.

Dobson D051 and Brewer B040 are independent from each other as one is not corrected to fit the ozone value of the other. The mean variation of the Brewer B040 during 2 years before and after an anomaly (Brewer B040 does not suffer from anomalies during these periods) is replicated on the same 4 years of Dobson D051. In that sense, the long-term trends of Dobson D051 and Brewer B040 stay independent. We add this consideration in the description of the MCH homogenization (section 3.1) in order to prevent any misunderstanding in the description of Figure 8.

The Dobson D051 and the Brewers stay independent from each other as one is not corrected to fit the ozone values of the others. Only the mean variation of the Brewers datasets during 2 years before and after an anomaly (Brewers data records do not suffer from anomalies during these periods) is replicated on the same 4 years of Dobson D051, allowing the long-term ozone variations to stay independent.

*To be frank, I would recommend a complete rewriting of the text. The authors should first decide the logical pathway they would like the reader to follow and then provide clear signposts and guidance to enable this.*

We changed the title, rewrote part of the abstract and modified the introduction in order to make the scope of the paper clearer. We reorganized the contents of the paper and added introductions to the sections and subsections. We followed the list of technical comments and modified the text in the corresponding sections in order to provide a red thread for the reader to follow. Further we think after clarifying that the NOAA work is not part of this paper, the reader should be able to follow well.

We do not list here all the modifications related to this comment but refer the reader directly to the modified version of the manuscript. As an example: the introduction to section 3:

**3. Homogenizations of the Dobson D051 dataset**

As the quality of a dataset is essential in order to estimate reliable long-term trends with uncertainties as reduced as possible, we first investigate the Arosa/Davos longest Umkehr ozone profile dataset and proceed to its detailed homogenization.

The worldwide longest Umkehr ozone profile dataset was recently impacted by short term anomalies due to instrumental changes and technical issues. It has been homogenized by two simultaneous but independent studies, one by the principal investigator group of the Dobson D051 instrument (further called MCH homogenization) and one by the NOAA (further called NOAA homogenization). Both homogenizations are described in sections 3.1 and 3.2 and compared in section 3.3. Details are provided in this work for the MCH homogenization while the reader is referred to Garane et al. (2022) and Petropavlovskikh et al. (2022) for details on the NOAA homogenization.

*I note there is no discussion of the physical basis of the deduced trends at different heights or the implications for ozone recovery, but I think that is quite reasonable given the scope is already very large.*

*Only the statistical uncertainty of the trends are provided. Clearly the overall uncertainties are much greater than these given the different results produced by the different instruments, the different homogenizations and the different trend analyses.*

The statistical uncertainty of a trend is derived from the standard deviation of the monthly means and the uncertainty related to the statistical regression method. This uncertainty is used in the

determination of the significance of the trend. The overall uncertainties of the trend can include an additional term accounting for the measurement uncertainties, including the step changes or other inhomogeneity of the D051 record (i.e. Bernet et al., 2019; Bernet et al., 2021). The overall uncertainty of the trends would indeed be larger. However, this estimate was not included in the trend analyses of the D051 record. We say in the conclusion that the trend disagreement between the two homogenized Dobson D051 datasets is likely related to the remaining differences left by the two homogenizations.

We added in the text of section 4.2:
Note that only statistical uncertainties are given in the paper, which allows to determine the significance of the trends. In order to check the agreement of trends derived from different datasets, uncertainties including a term accounting for remaining steps and inhomogeneities in the dataset should be considered.

**Specific comments**

*The title is too general, as it implies all Umkehr datasets from Dobsons and Brewers around the world are going to be considered rather than those from just one location. The acronym "DLM" should be expanded.*
We modified the title as suggested.
Dynamic Linear Modeling estimates of long-term ozone trends from homogenized Dobson Umkehr profiles at Arosa/Davos, Switzerland.

*Line 6 – You talk about there being six instruments available but then go straight to D051, without explaining why this one of the six is the focus.*
Yes, the referee is right. This was not clear. We have amended the text accordingly.

In Arosa/Davos, the Dobson D051 is the station's primary instrument for continuous Umkehr profile time serie. It was dedicated exclusively to Umkehr measurement from 1988 until February 2013, when total ozone measurement was added to the schedule. The number of observations dedicated to Umkehr was not impacted and the number of retrieved Dobson D051 Umkehr profiles was kept to two profiles per day up to now. This frequency in observations allows the computation of statistically reliable monthly means for trend estimations. However, the instrument operations recently suffered from anomalies following technical interventions. Therefore, a complete homogenization of the Dobson D051 Umkehr data record has been performed and is described in this paper.

*Lines 7-11 This sounds like the main goal of the work was to homogenize Dobson 051 using the co-located Brewers as a reference. However in other places (eg the title) the implication is you are calculating trends from all available instruments. I would prefer the abstract provide much clearer guidance to the reader about what the main idea of your work is.*
We modified the abstract in order to emphasize the homogenization work rather than the assessment of trends.

Six collocated spectrophotometers based in Arosa/Davos, Switzerland, have been measuring ozone profiles continuously since 1956 for the oldest Dobson instrument and since 2005 for the Brewer instruments. The datasets of these two ground-based triads (three Dobsons and three Brewers) allow continuous intercomparisons and derivation of long-term trend estimates. Mainly, two periods in the post-2000 Dobson D051 dataset show anomalies when compared to the Brewer triad time series: in 2011-2013, an offset has been attributed to technical interventions during the renewal of the

spectrophotometer acquisition system, and in 2018, an offset with respect to the Brewer triad has been detected following an instrumental change on the spectrophotometer wedge.

In this study, the worldwide longest Umkehr dataset (1956-2020) is carefully homogenized using collocated and simultaneous Dobson and Brewer measurements. A recently published report (Garane et al., 2021) described results of an independent homogenization of the same dataset performed by comparison to the Modern-Era Retrospective analysis for Research and Applications version 2 (MERRA-2) Global Modeling Initiative (M2GMI) model simulations. In this paper, the two versions of homogenized Dobson D051 records are inter-compared to analyze residual differences found during the correction periods. The Aura Microwave Limb Sounder (MLS) station overpass record (2005-2020) is used as an independent reference for the comparisons. The two homogenized data records show common correction periods, except for the 2017-2018 period, and the corrections are similar in magnitude.

In addition, the post-2000 ozone profile trends are estimated from the two homogenized Dobson D051 time series by Dynamical Linear Modeling (DLM) and results are compared with the DLM trends derived from the colocated Brewer Umkehr time series. By first investigating the long-term Dobson ozone record for trends using the well-established multi-linear regression (MLR) method, we find that the trends obtained by both MLR and DLM techniques are similar within their uncertainty ranges in the upper and middle stratosphere but that the trend's significances differ in the lower stratosphere. Post-2000 DLM trend estimates show a positive trend of 0.2 to 0.5 %/year above 35 km, significant for Dobson D051 but lower and therefore non significantly different from zero at the 95% level of confidence for Brewer B040. As shown for the Dobson D051 data record, the trend seems to become significantly positive only in 2004. Moreover, a persistent negative trend is estimated in the middle stratosphere between 25 and 30 km. In the lower stratosphere, the trend is negative at 20 km with different levels of significance depending on the period and on the dataset.

*Line 8, 9 – I am not sure "technical intervention" is really the best term, I think something like "instrumental changes" would be more accurate.*
Ok , terms have been adapted.

in 2018, an offset with respect to the Brewer triad has been detected following an instrumental change on the spectrophotometer wedge.

*Line 11 Throughout the manuscript, it is not clear to me what exactly you mean by "OEM" – do you mean an optimal estimation method (ie Rodgers) or the particular optimal estimation method as implemented in the standard Brewer algorithm (ie Petropavlovskikh)?*
By "OEM" we mean the optimal estimation method implemented by Petropavlovskikh in the standard Dobson and Brewer algorithms, which is an OEM and is based on Rodger's method. The OEM acronym is now defined and we add a reference to Petropavlovskikh et al. 2005a or Stone et al. 2015 when the term OEM is used.

The continuous and automated measurements (2 min cycle) are interpolated to 12 nominal SZAs and profiles are retrieved from ground to 50 km using Optimal Estimation Method (OEM) (Rodgers, 2000) implemented in Petropavlovskikh et al., 2005a.

and

The Umkehr ozone profile can be retrieved from three measured wavelength pairs (McElroy et al. 1995, Stone et al., 2015) by OEM.

*Lines 12-14 You should explain better to the reader how the alternative NOOA homogenization fits in – is it part of the scope of this work or is it just going to be used for comparison?*

We modified the abstract to make it clear that the NOAA homogenization is not part of this work and will be used for comparison purposes.

A recently published report (Garane et al, 2022) described results of independent homogenization of Arosa Umkehr  record that was performed by means of the Modern-Era Retrospective analysis for Research and Applications version 2 (MERRA-2) Global Modeling Initiative (M2GMI) model simulations. In this paper, the two versions of homogenized Dobson 051 records are inter-compared to analyse residual differences found during the correction periods. The Aura MLS station overpass record (2005-2020) is used as an idependent reference for the comparisons.

*Line 15 MLR trends were also calculated.*

MLR trends estimation is now mentioned in the abstract.

By first investigating the long-term Dobson ozone record for trends using the well-established multi-linear regression (MLR) method, we find that the trends obtained by both MLR and DLM techniques are similar within their uncertainty ranges in the upper and middle stratosphere but that the trend's significances differ in the lower stratosphere.

*Line 20 As a comment, the fact the trends depend on the dataset raises the interesting question of their true significance.*

Indeed, the question of the significance of a trend is justified and that is why we compare our results with other published results. A trend can be significantly different from zero and two trends can be significantly different from each other. Here we refer to the fact that a DLM trend may be significantly different from zero for a limited period of the time range of the dataset, the trend value and the significance of the trend vary within the time range of the data record. This is an intrinsic characteristic of the method and does not question the significance of a linear trend estimated from the same dataset. Moreover, trends estimated from two different datasets can agree within their uncertainties without excluding one of them being significantly different from zero and the other not.

*Lines 22-63 You should also include the findings of Godin-Beekmann et al. 2022 here. This is a very important reference for this discussion.*

Yes, the referee is right. The findings of LOTUS2 are now included in the introduction.

Upper stratospheric post-2000 ozone trends are reported to be significantly positive in the three broad latitude bands, with values of ~2.2+-0.7% per decade at 2.1 hPa in the NH, while non-significant negative ozone trends are derived in the lowermost stratosphere, with however large uncertainties (Godin-Beekmann et al., 2022).

and

The Umkehr data records are still extensively used for trend estimates along with datasets from other ground based techniques, satellites and models (Steinbrecht et al., 2017; Harris et al., 2015; Petropavlovskikh et al., 2019; Tarasick et al., 2019; Godin-Beekmann et al., 2022).

*Lines 29-63 The tenses are inconsistent, you have Chipperfield 'pointed to', but Wargan 'confirms',  and also 'have been reported' (Bognar)*

The tenses have been adapted. Present simple and present in reported speech are now used throughout the section.

Chipperfield et al. (2018) points to …

Orbe et al. (2020) associates …

Non monotonic post-2000 trends are also reported in Arosio et al. (2019) …

More recently, DLM trend estimates on SOS (SAGEII, Osiris (Optical Spectrograph and InfraRed Imaging System) and SAGEIII) merged satellite data record are reported (Bognar et al., 2022) and indicate …

*Line 40 You say MLR has been used for "ozone trend estimation" but the discussion that follows seems to be only about ozone profiles and not total ozone. You should either give references for the use of MLR in total ozone trend estimation (where of course it has also been widely used) or say "vertically resolved ozone trend estimation" or words to that effect.*

We do not want to discuss total ozone trends here, therefore we changed "ozone trend estimation" into "vertically resolved ozone trend estimation".

Multilinear regression (MLR) is widely and consistently used for vertically resolved ozone trend estimation.

*Line 65 Ozonesondes, not radiosondes (this occurs later too at line 166)*
"Radiosondes" has been replaced with "ozonesondes" here and at L166.

Beginning in 1956 for the oldest, the Umkehr records were unique at that time since satellites records only became available in 1979 (McPeters et al., 1996b; Bhartia et al., 2013) and ozonesondes, starting in 1960 (Smit et al., 2007), do not reach the upper stratosphere.

*64-66 Not just that, I would have thought there would be many Umkehr records at stations for which there weren't ozonesondes at the time, or in fact, where ozonesondes have never been flown.*
Yes, that's true, Umkehr records could have been used when and where the ozonesondes were not available.
We prefer a general consideration first on the existence of Umkehr, satellites and ozonesonde measurements technique in time and secondly in their capacity to measure in the upper stratosphere, we do not mean considering specific stations having or not the possibility to use the existing techniques.

*Lines 76-78 Do you mean that Umkehr records only exist in the Northern Hemisphere mid- latitudes? (And not the Arctic, Antarctic, tropics or southern-hemisphere?) What about the Umkehr results shown in Godin-Beekmann et al. 2022?*

We do not mean that Umkehr records only exist in the Northern Hemisphere mid- latitudes, however, throughout the manuscript, we mention the trends values only for the NH for comparison purposes with the trend estimated at the Arosa/Davos station which lies in this latitude band.
We added references to outer Northern Hemisphere stations with available Umkehr data record in the introduction.

Dobson Umkehr ozone profile data records, which are distributed all around the world (Petropavlovskikh et al., 2022; Godin-Beekmann et al., 2022; Stone et al., 2015; Miyagawa et al., 2009; Garane et al., 2022), have been extensively used in the pre-1998 stratospheric trend estimates (Reinsel et al., 1989; Randel et al., 1999; Miller et al., 1995) .

*Lines 80-81 I don't think that can be right – I have listed at least one example in the references (Fitzka et al.) Perhaps I have misunderstood.*

We would like to thank the referee for bringing this reference to our attention. The linear trends reported here are estimated with the Senn's Q method and significances are assessed with the Mann-Kendall test, both techniques exclude the use of explanatory variables. Same comment about the explanatory variables can be done for Fragkos et al, 2018 (which is not a peer reviewed publication but a EGU presentation). Trend estimation based on Brewer datasets including the use of explanatory variables and by DLM have not been published yet. We rephrased for clarity, and add Fitzka et al. in the references list.

However, trend estimations on Brewer Umkehr data records are sparse. A study using simple linear regression, without consideration of explanatory variables, applied to data from the Brewer 005 of Thessaloniki presented by Fragkos et al. (2018) reports 1997-2017 statistically significant positive trends above 35 km of 0.3%/year and non statistically significant trends below. Fitzka et al. reports on linear trends estimated with the Senn's Q method and significances assessed with the Mann-Kendall test. We innovate here by estimating Brewer Umkehr trends considering explanatory variables in the regression by DLM.

*Lines 81-83 This sentence seems to contradict the previous one – re-word to make the distinction clear.*
See response to the previous comment.

A study using simple linear regression, without consideration of explanatory variables, applied to data from the Brewer 005 of Thessaloniki presented by Fragkos et al. (2018) reports 1997-2017 statistically significant positive trends above 35 km of 0.3%/year and non statistically significant trends below.

*Line 91 This makes it sound as if the NOOA homogenization is not part of this work but is being used as a comparison – in other places quite a different impression is given*
The NOAA homogenization is not part of this work and we modified the manuscript to make it clear in this regard. The NOAA homogenization has been described in Petropavlovskikh et al, 2022 and Garane et al, 2022 in much details and is only compared to the MCH homogenization here (see the response in the general comment section of this report).

A recently published report (Garane et al., 2021) described results of an independent homogenization of the same dataset performed by comparison to the Modern-Era Retrospective analysis for Research and Applications version 2 (MERRA-2) Global Modeling Initiative (M2GMI) model simulations.

and

In parallel but in a separate work, a homogenization and a correction for the stray light effect of the same Dobson dataset has been performed by NOAA (Garane et al., 2021; Petropavlovskikh et al, 2022).

*Line 96 I would suggest starting with a very brief (one sentence) introduction of what an Umkehr observation is before getting into all the details.*
We would like to thank the referee for the suggestion, an introductory sentence has been added.

The Umkehr technique allows low-resolution retrieval of ozone profiles from measurements made by Dobson and Brewer spectrophotometers.

*Line 98 "relocation" rather than "relocalisation"*
Thank you, done.

The progressive relocation of the Dobson and Brewer triads from Arosa to Davos ….

*Lines 100-105 , 109-11 Would it be possible to re-write this section to make it easier for the reader to follow? Perhaps a table would help?*
As suggested by both reviewers, we added a table in section 2.1. We believe that with the table the section becomes sufficiently clear.

| Instrument | | Time range | Time resolution |
|---|---|---|---|
| Dobson | D015 | 1956-1988 | 2 profiles/day |
| | D051 | 1988-now | 2 profiles/day |
| | D062 | 1998-now | 4-6 profiles/month |
| | D101 | 1988-now | 4-6 profiles/month |
| Brewer | B040 | 1988-now | 2 profiles/day |
| | B072 | 2005-now | 2 profiles/day |
| | B156 | 2005-now | 2 profiles/day |

**Table 1.** Time ranges and time resolutions of the Dobson and Brewer Umkehr measurements at the Arosa/Davos station.

*Line 102 Is this the last we hear of D015?*

A description of the homogenization performed at the D015 to D051 transition is now added at the beginning of section 3.1.

The Arosa/Davos Umkehr time series is composed of Dobson D015 measurements from 1956 to 1988 and Dobson D051 since then. The quality of the homogenization of the Dobson D015 to Dobson D051 transition has been ensured by one year of parallel measurements (1988) allowing an adaptation of the D015 N values to the D051 N values. For each SZA, the 1988 mean difference between the D051 and the D015 N values has been added to the D015 values. The 1956-1987 ozone profiles have then been retrieved from the Dobson D015 corrected N values. No statistical correction has been performed on the D015 data record.
We report here about the complete homogenization of the 1988-2020 Umkehr Dobson D051 time series by…

*Lines 118-119 Considering the different stray light characteristics of single and double monochromator Brewers, how well do their Umkehr results agree with each other?*
The level of stray light rejections differs between single and double monochromator Brewers, where double monochromator Brewers have a significantly reduced contribution from stray light. The stray light is a function of SZA. Therefore there will be a constant bias between profiles retrieved with different levels of the stray light. We use here the difference of the Dobson N values to a mean of the 3 Brewer N values during a limited period of time (2years at a time). As the 3 instruments did not change during these periods, we can consider that the stray light characteristics of the single and the double monochromator Brewers did not change either during these periods, and are therefore not influencing the ozone variation during these limited periods.

*Lines 130-135 I think your description of the Umkehr method could be made clearer. Stone et al. 2015 did a good job. If the shorter wavelength is scattered above the ozone layer, wouldn't its intensity*

*decrease due to passing through more ozone?*

We have modified our description of the Umkehr method based on the description of Stone et al, 2015 and added Stone et al, 2015 as a reference.

The Umkehr method is based on the measurement of the ratio of downward scattered zenith sky radiation for two wavelengths in the UVB-UVA range from 300 nm to 330 nm (Huggins absorption band) which are subject to different strengths of ozone absorption, the shorter wavelength being more strongly absorbed by ozone. This ratio changes as a function of SZA during sunset and sunrise due to changes in the scattering height along the zenith (Mateer, 1965; Stone et al., 2015). As the SZA is increasing from 60° to 90°, the scattering height is increasing, and the two intensities decrease because of increased absorption and scattering by ozone and air molecules. As the shorter wavelength has a higher scattering point than the longer wavelength, its intensity is decreasing faster than the longer wavelength intensity as long as both scattering heights are below the ozone maximum. At high SZA, the scattering height for the shorter wavelength is above the ozone maximum and the scattering height of the longer wavelength is still below the ozone maximum. The shorter wavelength intensity decreases then less rapidly than the longer wavelength intensity and the ratio reaches a maximum at high SZA called the Umkehr effect (Götz et al., 1934).

*Line 135 You could reference Götz as the originator*
Yes, the referee is right. A reference to Götz et al, 1934 has been added.

the ratio reaches a maximum at high SZA called the Umkehr effect (Götz et al., 1934).

*Lines 144-148 The reader can't assess from this description whether the empirical correction is a good idea or not. Do you think it has any significant effect on the trends? Presumably it hasn't been used the whole time since 1956?*
This empirical correction has effectively been used since the beginning of the data record, is currently under investigation/modification/automatisation and will be the subject of a future publication. It has been checked with MLR that the effect of small cloud corrections on the trend is negligible. The purpose here is only to describe the Dobson measurements and the applied quality control. We mention now that a small cloud correction of the N values does not influence the ozone profile trend significantly.

It was shown that the effect of small cloud corrections of the N values on the vertically resolved ozone trends is negligible.

*Line 154 "is commonly retrieved" – from this wording it is not clear if you are talking about what other people do, or what is being done here.*
By "commonly retrieved" we refer to the Brewer instrument standard built-in retrieval which uses the "short" Umkehr method and all measured wavelengths. The reference to this retrieval is quite old (McElroy, 1995, https://doi.org/10.1029/94JD03250) and the retrieval does not seem to be used in the EUBREWnet community. Therefore, we reword and specify what is being done here.

The Umkehr ozone profile can be retrieved from three measured wavelength pairs (McElroy and Kerr, 1995, Stone et al., 2015) by OEM. For similarity with the Dobson Umkehr measurement, the intensity ratio of only two wavelengths are used here: …

*Line 165 Does it matter that the a priori profiles are now 20 years out of date, and ozone has increased*

*since then in the upper levels, as your work shows?*

We use the retrieval of Petropavlovskikh et al, 2005 which has been optimized with no constrain of the a priori to total ozone column and for reduction of the a priori influence on the derived trends through the a priori covariance matrix determination. We use only the portion of the Umkehr profile where the a priori influence is negligible. This altitude range is determined based on the averaging kernels. It is therefore safe to say that the a priori does not influence our trend results. Further, the a priori should be representative for the entire period under consideration. Since our data set goes back to the 1950s we believe our choice of the a priori is fine.

*Lines 169-170 Information below level 4 "is not independent" – independent of what? Each other?*
This has been reworded for better clarity.

In the layers below Dobson Layer (DL) 4, peaking at 20 km, for both instruments, the Averaging Kernels (AKs, not shown) show sensitivity of observations to ozone variability in several layers, and therefore the partitioning of the retrieved ozone in individual layers is based on the a priori information.

*Lines 170-171 You say a generic stray light correction "can be applied" – but have you applied it? Is it different for Dobsons and Brewers? Does it affect the results?*

We did not apply the generic stray light correction neither on Dobsons (as said on P11 L269) nor on Brewers data retrieved by MCH. But, yes the stray light correction could be different in Dobson and Brewers (and it is larger in single monochromator Brewers than in double monochromator Brewers) and NOAA version of the Dobson RT uses standardized corrections to minimize seasonal biases in the retrieved ozone. The step changes in the Umkehr record can be related to a change in the amounts of stray light. This is taken into account when corrections are iteratively adjusted by NOAA after the step detection. Petropavlovskikh et al., 2011 shows that this generic correction affects the ozone values but does not influence the trends as long as it is not varying with time.

A generic stray light correction can be applied to reduce systematic biases in the Dobson Umkehr retrieved profiles (Petropavlovskikh et al., 2011). The NOAA version of the Dobson retrieval applies this correction while the MeteoSwiss (MCH) version does not. The seasonal bias between Dobson and Brewers is reduced when a stray light correction is applied to the Dobson record Petropavlovskikh et al., 2009. Moreover, as a step change in the record can be related to a change in the amount of stray light, a proper correction of the stray light effect can help to reduce the magnitude of the step.

*Lines 175-184 I am very confused about why this section is here – it seems completely irrelevant.*
This section has been removed. Same for P18 L386-390.

*Lines 187-189 The way this is written it seems to contradict itself – you say "in this study we use ozone profiles … given on 55 pressure levels " but then you "only consider" 10-75 km - please reword for better clarity.*
The referee is right. This is now reworded.

Ozone profiles from the version 4.2 dataset are given on 55 pressure levels from 1000 to 1e-5 hPa (Livesey et al., 2018). However, the useful vertical range for Aura MLS ozone leads us to only consider Aura MLS data from 10 to 75 km (in this range, the Aura MLS vertical resolution is about 2.5 to 4 km) for Aura MLS overpasses above Arosa (±3° in latitude and ±5° in longitude).

*Line 190 I assume this is the latitude and longitude of Arosa (and not just "Switzerland") but it doesn't seem to be actually given anywhere in the manuscript.*

Yes, correct. The latitude and altitude of Arosa are now given in section 2.1.

TCO and ozone profile measurements with Dobson (and Brewer) spectrophotometers were performed at Arosa (46.82° N, 6.95° E) from 1926 (and 1988) to 2021 and at Davos since 2012.

and "Switzerland" is replaced with "Arosa"

Aura MLS overpasses above Arosa (±3∘ in latitude and ±5∘ in longitude).

*Line 191 It don't think it's reasonable to cite Ziemke (2017) for this calculation – it appears in Godson (1962) in his discussion of Umkehrs.*

Reference to Ziemke et al., 2017 has been replaced by reference to Godson, 1962 as suggested.

These ozone profiles are interpolated on the Umkehr pressure levels $p_i$ and converted to DU following Godson (1962):

*Line 194 – The implication is Petropavlovskikh et al. 2022 defined the definitions of the Umkehr layers.*

This is now reworded.

Approximative heights are given as in Petropavlovskikh et al., 2022.

*Line 196 I don't think 'the 2008 homogenization' has been mentioned before – what are you talking about?*

As the 1988 D015 to D051 homogenization has been reprocessed for this study, the 2008 homogenization is finally exclusively a reprocessing of the N values with adapted shaft encoder positioning. This was not published but reported as an internal report. We remove any mention of the 2008 "homogenization" as it may not be considered as a homogenization but only as a reprocessing and we discuss in more details the correction of the D015 to D051 transition in section 3.

The Arosa/Davos Umkehr timeseries is formed by Dobson D015 measurements from 1956 to 1988 and Dobson D051 since then. The quality of Dobson D015 to Dobson D051 transition has been ensured by one year of parallel measurements (1988) allowing an adaptation of the D015 N values to the D051 N values. For each SZA, the 1988 mean difference between the D051 and the D015 N values has been added to the D015 values. The 1956-1987 ozone profiles have then been retrieved from the Dobson D015 corrected N values. No statistical correction has been performed on the D015 data record. We report here about the complete homogenization of the 1988-2020 Umkehr Dobson D051 time series by…

*Line 201 If you use the wording "a technical issue in the metadata", it sounds like there is a problem with the metadata. I think it would be better to say something like "an instrumental change recorded in the metadata".*

Ok, done.

However, a correction is only applied if it correlates with a technical issue reported in the metadata.

*Line 205 "retrieval iterations higher than 3" has not been explained.*

We reword line 205 and explain the role of the number of iteration in section 2.1.4.

Only simultaneous measurements, not flagged for bad weather conditions, volcanic eruptions, and number of iterations, are considered.

and

The quality check of the retrieved ozone profile includes assessment of the number of iterations (fewer than four is considered a good profile) and the condition that the difference between observed and retrieved Umkehr observations at all SZAs remains within measurement uncertainty (Petropavlovskikh et al., 2022).

*Lines 205-208 Given there are small but identifiable differences between the Dobson and the Brewer, does this limit the effectiveness of your approach?*

As Dobson and Brewer ozone profiles are not identical, we expect small differences between the Dobson and the Brewer N values but we expect this difference to be constant from year to year for each ozone layer. The annual variation of the differences is not a problem here as we represent the deseasonalised anomalies. The approach is limited by the remaining variability of the difference but we consider changes when they are larger than the standard deviation of the Brewer Dobson differences.

Note that the annual cycle is not visible on the representation of deseasonalised anomalies as in Figure 2 and that we consider changes when they are larger than the standard deviation of the Brewer Dobson differences.

*Line 210 It sounds like good metadata is only available from 2000 onwards – does this affect the confidence of the trends for earlier periods (eg Figure 4)?*
Metadata are good and reliable from the beginning of the measurements. In this paper, we decided to focus on the post-2000 period first because our homogenization technique implies collocated and redundant measurements which are only available since 1988/1998/2005 and secondly because the post-2000 trends are a key parameter of the ozone recovery studies. The Arosa/Davos pre-2000 trends rely on the Dobsons D051(1956-2000) and D101(1988-2000) and have not been investigated in this study. We reworded the first sentence of the paragraph to avoid confusion concerning the quality of the metadata.

If we focus on the post-2000 period, where several collocated and redundant measurements are available, systematic anomalies of the Dobson D051 are noticed.

*Lines 214-215 This effect also seems evident in 2009?*
The magnitude and the systematic character of the anomaly are to consider. In 2009, the magnitude of the anomaly is much lower than in 2010 and can be considered as part of a remaining variability in the difference between Dobson and Brewer ozone profiles. We add the consideration of the magnitude of the anomaly in the text.
Note that the annual cycle is not visible on the representation of deseasonalized anomalies as in Figure 2 and that we consider changes when they are larger than the standard deviation of the Brewer Dobson differences.

*Lines 216-217 It only looks lower in the lower altitude levels ?*

Yes, correct and in the upper altitude levels. We mention in the text only the main feature of the difference to highlight the period and rely on the plot for the rest.

*Line 218 But not at the level around 30 km ?*

Yes correct. We choose to describe only the main differences to highlight them. The details of the difference can be seen on the plots.

*Lines 219-221 It seems a shame that the observing routines changed in this way – was there a reason for the change?*

Yes, the referee is right, it's a shame. First, this period should have been removed in Figure 2 and has unfortunately not been in the last version of the figure. This is an error. The figure has now been corrected. Second, the 2014 anomaly should be considered with caution because of the very reduced number of measurements at that period (and effect of the moving average in the plot). Many data are missing during this technical and staff transition period or have to be flagged because of roof opening issues. It is therefore very difficult to investigate this period.
After 2018, the Umkehr measurement with D062 and D101 have been drastically reduced as priority has been given to total ozone measurements.

The comparison of Dobson D051 with the collocated Dobsons around 2014 and after 2018 are to be taken with caution due to the very limited number of measurements of Dobson D051 in 2014 and of Dobson D062 and Dobson D101 during these periods. Around 2014 (technical and staff transition period), many data are missing or have to be flagged because of roof opening issues. After 2018, the Umkehr measurement by Dobson D062 and Dobson D101 have been drastically reduced as priority has been given to total ozone measurements.

*Lines 222 I found the table somewhat hard to follow and perhaps therefore slightly unconvincing. I suggest at least considering whether it would be better to relate the changes in the instrument to the observed discrepancies using text. At present it is hard for the reader to assess what you've done and its validity.*

The purpose of Table 1 is to provide explanations for steps and anomalies in the data and hence links primarily to the black frames in Figure 2. The table provides a lot of information and is therefore both important and a bit hard to digest. We believe the table is important to understand our considerations in the paper. It would be very heavy to describe every instrumental change or issue in the text of section 3.1 only. However, we refer to the table in the text each time a correction is mentioned and hope this will help to assess what is done and why. We modified the column captions and add comments in the last column of the table to assess the validity of an effect of the change on the time series. We modified the table caption.

| year of Dobson D051 anomaly | Technical issue/instrumental change | Homogenized period | Time range used for the offset determination | Redundant datasets used for the offset determination | Comment |
|---|---|---|---|---|---|
| 1988 | D015 to D051 | Before 1988.01.01 | 1987.01.01 - 1988.01.01 and 1988.01.01 - 1989.01.01 | D015 and D051 simultaneous measurements | Instrumental change: Dobson D051 replaces Dobson D015. Adjustement of the dataset measured by D015 before 1988 to the dataset measured by D051 after 1988 |
| 2003 | Intercomparison and new RtoN table New RtoN table considered | Before 2003.07.19 | 2001.07.19 - 2003.07.19 and 2003.07.19 - 2005.07.19 | D062 and D101 mean values | Adjustement of the optics during the IC. Remaining inhomogeneity despite the use of a new RtoN table |
| 2010 | - | 2010.01.01 - 2010.06.30 | - | - | Does not correspond to any technical issue. Period limited to 6 month. Not corrected. |
| 2011-2013 | New electronics (2011.03.21) New Qlever motors (2012.02.15) New software 3V3 (2013.03.26) | 2011.04.01 - 2013.04.01 | 2009.04.01 - 2011.04.01 and 2015.04.01 - 2017.04.01 | B040, B072 and B156 mean values | 2014 not considered (number of measurement low and problematic period). Refurbishment of the electronics (HV, motors, feedback loop, amplification board) and position of Q2-lever as function of the room Temperature. Q-lever motor are essential is the selection of the wavelengths. |
| 2018 | New wedge steel band (2018.05.06) IC(2018.08.07-17): adjustments on optics AROSA to DAVOS (2018.09.28) | Before 2018.05.01 | 2016.05.01 - 2018.05.01 and 2018.05.01 - 2020.05.01 | B040, B072 and B156 mean values | The optical attenuator consists of a moving neutral-density filter (the optical "wedge") attached to a graduated rotating disc (R dial). The wavelength pair selection is achieved by rotating a pair of quartz plates (Q1 lever, Q2 lever) through which the light beam passes. |

**Table 2.** Dobson D051 homogenization description: determined time of Dobson D051 anomaly, technical issues or instrumental change which is considered as the source of the anomaly, homogenized period, time ranges for the offset calculation, used redundant datasets for the offset calculation, and details of the technical issue

Table 2 summarizes the Dobson D051 problematic periods, the technical issue reported at these periods, and the time ranges and redundant datasets used for the offsets determination.

For each period that requires a correction (see Table 2) we apply to the N values a SZA dependent offset which is constant over the period to be corrected.

*Lines 228-229 Given the Brewer uses different wavelengths, with a different assumed stratospheric temperature, and has different stray light characteristics at high SZA, is it reasonable to apply a constant offset to the N values?*
The offset is constant in time and depends of the SZA. We can reasonably assume that the "Brewer different wavelengths, the different assumed stratospheric temperature, and the stray light characteristics at high SZA," do not change with time during the considered 2 years limited periods.

*Line 244 You need to include some introduction to this section rather than going straight into the details, explaining in broad terms how this part fits into the overall picture of what you're doing, and secondly, how this homogenization is different to yours in its basic approach.*
We subdivide now section 3 in three subsections "3.1 MCH Homogenization of the Dobson D051 dataset", "3.2 NOAA homogenization of the Dobson D051 dataset", and "3.3 Comparison of the homogenizations of the Dobson D051 dataset". As suggested, we added an introduction to what is now subsection 3.2.

In parallel but in a separate work, a homogenization and a correction for the stray light effect of the same Dobson dataset has been performed by NOAA (Garane et al., 2021; Petropavlovskikh et al., 2022). They use the comparison of the Dobson D051 dataset with the M2GMI model on the N values level when the MCH homogenization uses the comparison with N values of the collocated instruments.

*Line 244 The implication is you do not use the 'correction for the stray light effect' – does that matter?*
See response to specific comment "Lines 170-171".

*Lines 249-250 I don't follow this. What is the difference between using a reanalysis and using specified dynamics? What are the specified dynamics based on if not the reanalysis?*

We tried to summarize a maximum. It looks like we actually added details without much explanation. The meteorology assimilated in the model is different: MERRA-2 on one side and SD simulations on the other i.e. the MERRA-2 assimilated meteorological fields are used by the model to simulate meteorology that is continuously adjusted to the MERRA-2 winds, temperature, and surface pressure.

We refer to Petropavlovskikh et al. 2022 (section 2.4) for details on P10 L246 and we rephrase the beginning of the paragraph for better clarity.

The NASA Global Modeling Initiative chemistry transport model (GMI CTM, Orbe et al., 2017; Wargan et al., 2018) is a full general circulation model that is driven by MERRA2 meteorological reanalysis throught the replay method (Gelaro et al., 2017). The simulation of the meteorological fields in the M2GMI model is continuously referenced against the MERRA-2 winds, temperature and surface pressure fields (Orbe et al., 2017). For the NOAA homogenization process, the M2GMI ozone and temperature profiles are selected for the Arosa station location.

*Line 251 "accounting" -> "accounting for"*

Typo has been corrected.

*Figure 3 – I think this diagram is very helpful – I like it.*

Thank you.

*Lines 257-258 Why would you try to reduce mean bias compared to AURA-MLS? The implication is MLS can be used as a reference to adjust the Umkehrs to. In that case why bother with the Umkehrs at all?*

See the answer to general comment "*As an example, from my understanding…*" on P3.
The NOAA homogenization method uses MLS as guidance to optimize the stray light correction (N-value correction) in the Umkehr retrievals. This correction reduces the bias between MLS and Umkehr but it does not change over time and therefore Umkehr record remains independent of MLS. We rephrase it for clarity.

An iterative modification of the N value correction is further performed for optimization of the stray light correction, adding a constant offset correction to the Umkehr dataset. This results in a reduced bias to other ozone records in the upper stratosphere but, as a constant offset, does not have any impact on the trends.

*Line 258 Why stop at 2018? This makes it seem as if the NOAA homogenization is a separate piece of work.*

Yes it is. As published in Petropavlovskikh et al, 2022. See now section 3.2.

In parallel but in a separate work, a homogenization and a correction for the stray light effect of the same Dobson dataset has been performed by NOAA (Garane et al., 2022; Petropavlovskikh et al., 2022).

*Line 261 "Our approach is different" – this strongly makes it seem as if the NOAA homogenization is a*

*separate piece of work.*
This has now been reworded.

The MCH homogenization approach is different in that the homogenization process aims to remove artificial steps in the Dobson D051 Umkehr profiles record while maintaining the constant offset between the datasets …

*Line 265 The term "MCH" has only appeared in a caption until now and should be properly explained*

Yes correct. "MCH" is now explained at its first occurence.

The NOAA version of the Dobson retrieval applies this correction while the MeteoSwiss (MCH) version does not.

*Lines 268-269 You don't say why this correction would cause high variability*

The straylight correction (Petropavlovskikh et al, 2009) affects differently the intensity measured at each SZA and is a function of this intensity. For the same SZA, the amount of correction is then different for each monthly mean value of the timeseries. A variability is then observed in the N value correction timeseries of Figure 4(a).
An explanation has been added.

The seasonal variability of the NOAA N values comes from the correction of observed N values for the stray light effect. Indeed, the straylight contribution varies with SZA and is proportional to the total column ozone value (Petropavlovskikh et al, 2009). For the same SZA, the amount of correction is different for each monthly mean value of the timeseries in proportion to the seasonal changes in total column ozone value (Fig. 4a).

*Line 283 I am not convinced the use of the abbreviations 'MS' and 'UpS' is beneficial, all things considered, but you might disagree.*
The abbreviations LS, MS and UpS have been replaced with lower stratosphere, middle stratosphere and upper stratosphere respectively.

*Lines 282-297 It seems to me that you are using AURA-MLS as a reference, in that you are judging the quality of the homogenizations by how well the data agrees with MLS. Is that what you really mean?*

We compare the two homogenizations of Dobson D051 by plotting their differences towards MLS. The difference of B040 towards MLS is also plotted. We do not compare the magnitude of the offset of the difference but we consider the variation of each respective offset with time in order to detect the eventual remaining issues with the timeseries. We refer to the comments and answers above ("As a prime example…" and "Lines 257-258"). Comparisons are important to gain confidence in the results. We do not think MLS is a reference in the sense that all data sets should look like MLS. But given the quality of MLS, to be in reasonable agreement with MLS and derived results is reassuring.

*Figure 5a – This plot is helpful but the black line is very hard to see. All four datasets seem to agree better prior to 2010.*
The difference in DL5 of all four datasets to MLS looks smaller before 2010. This has to be related to the fact that no corrections are applied on the Dobson D051 datasets between 2003 and 2010 but several corrections are applied after 2010. Moreover, the variability of the differences to MLS of each dataset is higher after 2010 while the mean values are constant. We mention this in the text now.

No clear offset in the difference to Aura MLS between the NOAA and the MCH homogenized record is reported in DL5. The variability of the differences to Aura MLS of each dataset looks higher after 2010 while the mean values are constant.

*Figure 5b Why is the black line only visible for a short period? Does it lie underneath the red line?*
Yes, the difference between the black and the red line is small.

*Figure 5c This is a very helpful plot.*
Thank you.

*Figure 5 Caption - You should state the heights or pressures corresponding to the layers DL5 and DL8 for the convenience of the reader.*
Heights and pressures can be looked up in Fig 1. We indicate now the approximate altitude for DL5 and DL8 in the figure titles.

*Line 296 Why wouldn't the model take into account the 'local variability'? Is the resolution too coarse?*

The spatial resolution of M2GMI is 0.625 degrees in longitude and 0.5 degrees in latitude. Therefore, it is possible that over the mountain areas with large ozone gradients, the model would have a hard time reproducing ozone variability in the troposphere. However, I would think that in the stratosphere, the processes are more homogenized so the M2GMI ozone profiles should be representative of stratospheric ozone variability. Nevertheless, it is possible that other atmospheric interferences (i.e. aerosols) can impact the Dobson readings of zenith sky radiance which would also impact Brewer observations, but might not be included in the M2GMI simulations. We amended the text by adding this last sentence.

As the MCH homogenization relies on the Brewer collocated datasets, it allows to take into account the local variability of the ozone DL8 content that the M2GMI model, base for the NOAA homogenization, probably does not consider. As the atmospheric processes are more homogenized in the stratosphere that in the troposphere, the M2GMI ozone profiles should be representative of stratospheric ozone variability. Nevertheless, it is possible that other atmospheric interferences (i.e. aerosols) can impact the Dobson readings of zenith sky radiance which would also impact Brewer observations, but might not be included in the M2GMI simulations.

*Lines 298-302 I found this hard to follow, sorry.*
We amended the explanation and try to make it as clear as possible without adding any figure to the manuscript. We however complete the response to the referee's comment with two figures. The first one (Fig A) shows the magnitude of the difference between the D051 homogenized records and the D051 as measured record in 2017. The second one (Fig B) shows that the NOAA homogenization method detects a change in the Umkehr ozone with respect to the M2GMI record starting in 2017.

Due to the occurence of an anomaly in 2018, which is particularly visible in DL8 for all datasets (Fig. 5(c)), the last correction applied to the dataset by the NOAA and the MCH homogenizations differ.
As the MCH homogenization considers a step correction in May 2018, the ozone increase during the 2018 anomaly is accounted for in the mean difference of the D051 dataset to the Brewers datasets of the pre- and the post-step periods. As a result, the calculated offset is small.

[Figure]

Figure A: DL5 and DL8 time series of the difference between the D051 homogenized records and the D051 as measured record.

The NOAA homogenization method detects a change in the Umkehr ozone with respect to the M2GMI record that starts a year earlier, in 2017.

[Figure]

Figure B: DL8 time series of the change in the Umkehr ozone with respect to the M2GMI record.

The ozone increase during the 2018 anomaly is accounted for only in the mean difference to M2GMI of the post-step period of the D051 dataset. Moreover, this post-step difference is overestimated as M2GMI doesn't seem to simulate any significant anomaly at that period. As a result, the calculated offset, applied in 2017, is probably overestimated.

*Lines 306 Why would you start the second trend in 2000? Is it appropriate for all heights? In line 331 you say it started in 1998.*
Yes correct, this is a residual from a previous text version. "2000" has been replaced by "1998" in the 4.1 and 4.2 sections. Trend plots and values are correct only the year had not been changed.

*Lines 308-311 and 323-324 Did you find that this selection of proxy variables gave good results? Did you need all of them?*
These are the classical standard proxies used for vertically resolved MLR trends (LOTUS: Petropavlovskikh et al., 2019 and Godin Beckmann et al., 2022, and refs therein), the same proxies were used for DLM to make trend comparisons consistent. All these proxies have been checked to be useful by investigating their coefficients and the residuals of the regressed timeseries.

*Line 309 The link is not for the QBO*

Yes correct. Modified.

https://www.geo.fu-berlin.de/met/ag/strat/produkte/qbo/index.html

*Line 310 When I follow this link, the data only seems to go back to 2004, not 1970 or 1956.*
Yes correct. Link modified.

https://www.iup.uni-bremen.de/UVSAT/Datasets/mgii

*Figure 6 Caption – You should explain in the caption what the blue and black lines are.*
Ok, mention of blue and black has been added in the caption.

(a-c) DLM (in blue) and MLR (in black) trend estimates in %/decade ± 2σ of Dobson D051 dataset for 3 DL between 20 and 40 km

*Lines 331-345 Would this be easier to represent in a table?*

We think that, as each value is given as text in the figure, a table would be redundant.

*Lines 348 Here, I think you should use the word "percentage" rather than just the symbol.*
"%" has been replaced by "percentage"

Note that the given DLM trend value in %/decade is an average of the percentage change per year

*Lines 360-372 Wouldn't it be more meaningful then just to calculate trends on the independent Umkehr levels?*

Yes, the referee is right. However, the issue here does not come from the dependent layers but from trying to draw conclusions for the whole LS and MS when only part of the DL2 is in the lower stratosphere and only part of the DL6 is in the middle stratosphere. We now draw conclusions at the DL level and not at the lower stratospheric or middle stratospheric level. The conclusions have been adapted.

In the middle stratosphere (DL5&6, 24–32 km), both homogenized records show a negative trend in DL5, persistent and significantly different from zero at the 95% confidence level since 2012 for the MCH homogenized Dobson D051 data record but slightly positive between 2002 and 2010 and non significantly different from zero at the 95% confidence level for the NOAA homogenized data record. In the lower stratosphere (LS, DL3&4, 14-24 km), the DL3 and DL4 trend estimates are non significantly negative before 1996 but significantly negative between 2008 and 2018 in DL4 for the MCH homogenized data record and non significantly negative for the NOAA homogenized Dobson D051 record.

*Lines 386-390 This information is presented without any context. Is the point that Boulder and OHP are also northern hemisphere mid-latitude sites?*
This paragraph has been removed (see specific comment "Lines 175-184").

*Lines 393-394 – "… has been homogenized on the observation data level" – what does that mean?*
The observation data are the N values by opposition to the ozone profile values which are the retrieved data. We replace the term with "raw data" which is a more commonly used term.

Data records of six collocated spectrophotometers were inter-compared on the raw data level (N values) and on the ozone profile level in order to detect anomalies.

**References**

*Fitzka, M., Hadzimustafic, J., and Simic, S. (2014), Total ozone and Umkehr observations at Hoher Sonnblick 1994–2011: Climatology and extreme events, J. Geophys. Res. Atmos., 119, 739– 752, doi:10.1002/2013JD021173.*

*Godson, W.L. (1962), The representation and analysis of vertical distributions of ozone. Q.J.R. Meteorol. Soc., 88: 220-232. https://doi.org/10.1002/qj.49708837703*

*Götz, F. W. P., A. R. Meetham, and G. M. B. Dobson, Proc. Roy. Soc. A 145, 416, 1934.*

*Stone, K., Tully, M. B., Rhodes, S. K., and Schofield, R.: A new Dobson Umkehr ozone profile retrieval method optimising information content and resolution, Atmos. Meas. Tech., 8, 1043–1053, https://doi.org/10.5194/amt-8-1043-2015, 2015.*

---

## Author Response (AR2)

**Response to the Editor request of a technical correction**:

*Dear Dr Maillard Barras and co-authors,*

*I am happy to accept your manuscript for final publication in ACP.*

*However, I have one request in terms of a technical correction: In Fig. 6a-c, I do not fully understand how the uncertainty ranges (i.e. shaded areas) are calculated. For MLR, for example, have you considered the uncertainty of the linear trend alone, or a combination between axis offset and trend? In terms of trend uncertainties, I would expect an uncertainty range growing with its temporal distance from a selected fixpoint (how was that chosen?), but what I see is an uncertainty range having the same width around the fit (which I would interpret as the uncertainty of the offset alone). Please could you describe in some more detail how the shaded ranges in Fig.6 a-c were calculated?*
*Except for this issue, your paper is fine and show go directly to the production.*

*Kind regards,*
*Gabriele Stiller*

Dear Editor,

Thank you for accepting our manuscript for final publication in ACP.

MLR trend uncertainties representations:

The MLR trend results are given as a difference in DU to the 1970–1980 and the 2000–2010 means (see section 4.1). The trend in DU per decade +/-2 sigma is calculated with respect to the reference. A slope of a corresponding DU per year is plotted and the +/-2 sigma constant value is reported along the slope.
However, if we represent the slope +/- its uncertainties, we get a higher (+ 2 sigma) and a smaller (-2 sigma) slope, forming a V shape from a starting point. This can be seen as a 0 DU/y uncertainty at the starting point which, in our case, is not the reference. That's why we plot the uncertainty using the constant 2 sigma offset.

By analogy with the representation of the DLM trend uncertainties (for which the uncertainty varies with the year as the trend is calculated and represented in DU/year), we reported the MLR trend per decade uncertainty on the DU/y trend line and assumed the uncertainty to be constant within the decade.

We added an explanation of what the shaded areas represent and how they are calculated at the beginning of section 4.3.

"The blue shaded areas show the non-constant 2 sigma uncertainties in DU/y estimated by the DLM. By analogy, for the MLR, the grey shaded areas report the uncertainty in DU/y calculated from the constant 2 sigma offset trend uncertainty in DU per decade."

Kind regards,

Eliane Maillard Barras (on behalf of all co-authors)